



Atmospheric
Chemistry
and Physics

# Technical note: Comparison and interconversion of pH based on different standard states for aerosol acidity characterization

**Shiguo Jia**[1,2], **Xuemei Wang**[3], **Qi Zhang**[1], **Sayantan Sarkar**[4], **Luolin Wu**[1], **Minjuan Huang**[1,2], **Jinpu Zhang**[5], and **Liming Yang**[6]

[1]School of Atmospheric Sciences, Sun Yat-sen University, Guangzhou 510275, P. R. China
[2]Guangdong Province Key Laboratory for Climate Change and Natural Disaster Studies, Sun Yat-sen University, Guangzhou 510275, P. R. China
[3]Institute for Environmental and Climate Research, Jinan University, Guangzhou 510632, P. R. China
[4]Department of Earth Sciences, Indian Institute of Science Education and Research – Kolkata, Nadia 741246, West Bengal, India
[5]Guangzhou Environmental Monitoring Center, Guangzhou 510030, P. R. China
[6]Department of Chemical and Biomolecular Engineering, National University of Singapore, Singapore 117576, Republic of Singapore

**Correspondence:** Xuemei Wang (eeswxm@sysu.edu.cn) and Liming Yang (cheylm@nus.edu.sg)

**Abstract.** Aerosol pH is often calculated based on different standard states thus making it inappropriate to compare aerosol acidity parameters derived thereby. However, such comparisons are routinely performed in the atmospheric science community. This study attempts to address this issue by comparing PM$_{2.5}$ aerosol pH based on different scales (molarity, molality and mole fraction) on the basis of theoretical considerations followed with a set of field data from Guangzhou, China as an example. The three most widely used thermodynamic models (E-AIM-IV, ISORROPIA-II, and AIOMFAC) are employed for the comparison. Established theory dictates that the difference between pH$_x$ (mole fraction based) and pH$_m$ (molality based) is always a constant (1.74, when the solvent is water) within a thermodynamic model regardless of aerosol property. In contrast, pH$_m$ and pH$_c$ (molarity based) are almost identical with a minor effect from temperature and pressure. However, when the activity coefficient is simplified as unity by thermodynamic models, the difference between pH$_m$ and pH$_c$ ranges from 0.11 to 0.25 pH units, depending on the chemical composition and the density of hygroscopic aerosol. Therefore, while evaluating aerosol acidity (especially, trend analysis) when the activity coefficient is simplified as 1, considering the pH scale is important. The application of this pH standardization protocol might influence some conclusions on aerosol acidity reported by past studies, and thus a clear definition of pH and a precise statement of thermodynamic model parameters are recommended to avoid bias when pH comparisons are made across studies.

## 1 Introduction

Aerosol acidity is of great scientific interest due to its effects on human health and atmospheric chemical processes (Amdur and Chen, 1989; Xue et al., 2011). Acidic aerosols are found to correlate with health effects including asthma, bronchitis, and others respiratory diseases along with reduced lung function (Amdur and Chen, 1989; Ricciardolo et al., 2004; Longo and Yang, 2008). Acidic aerosols can also contribute to the bioavailability of iron and phosphorus in open oceans (Nenes et al., 2011; Zhu et al., 1992) and acidic sea salts have the potential to catalyze halogens to deplete tropospheric ozone (O$_3$) (Keene et al., 1998; Pszenny et al., 2003; Simpson et al., 2007). Moreover, aerosol acidity plays a key role in the gas-particle partitioning of species such as HCl/Cl$^-$, HNO$_3$/NO$_3^-$ and NH$_3$/NH$_4^+$, and is thus vital for predicting lifetimes of gaseous compounds such as HCl, NH$_3$

**Published by Copernicus Publications on behalf of the European Geosciences Union.**

and HNO$_3$ in the atmosphere (Nemitz et al., 2004; Oss et al., 1998). Further, aerosol acidity is known to affect the formation of secondary organic aerosols (SOA); e.g., experimental studies show that seed aerosols with acidic surfaces can enhance the formation of organosulfate SOA upon reaction with volatile organic compounds such as octanal, carbonyls, isoprene, limonene, and caryophyllene (Jang et al., 2002).

The most accurate parameter to characterize aerosol acidity is considered to be pH. The other parameters often used as proxies of aerosol acidity do not offer information on how acidic the particles are when they are present as aqueous droplets (Pathak et al., 2004). For example, strong acidity (defined as nmol of total H$^+$ per m$^3$ of air measured in water extracts of particles using the USEPA Reference Method, USEPA, 1992) and ion charge balance are unable to distinguish between free and undissociated H$^+$ (e.g., protons associated with bisulfate) (Pathak et al., 2004; Hennigan et al., 2015). Ammonium-to-sulfate ratio and cation-to-anion ratio are unable to provide any measure of the degree of aerosol acidity even qualitatively (Hennigan et al., 2015). Lastly, free acidity (defined as the actual concentration of free H$^+$ per m$^3$ of air, not including the H$^+$ released from bisulfate ions in aqueous extracts) represents the quantity of H$^+$ in a specific volume of air while neglecting the concentration of H$^+$ in liquid water (Pathak et al., 2004).

As per the International Union of Pure and Applied Chemistry (IUPAC), pH is defined as the negative log (base 10) activity of hydrogen ions (https://goldbook.iupac.org/html/P/P04524.html TS3). It is immeasurable because its definition involves a single ion quantity, the hydrogen ion activity (Baucke, 2002). Therefore, the value of pH is not an absolute one but depends on either how it is measured or the model used to calculate it. Especially, for aerosol pH, a commonly accepted measurement method is lacking despite some recent developments (Rindelaub et al., 2016), and it is usually calculated from thermodynamic models in practice.

One issue in comparing aerosol pH across studies even when calculated using the same model in actual practice is that different standard states can be used while defining the activity of H$^+$ ions. Although it is recommended that pH be defined based on the standard state of 1 mol H$^+$ kg$^{-1}$ solvent (molality based) (https://goldbook.iupac.org/html/P/P04524.html), other standard states such as 1 mol H$^+$ dm$^{-1}$ solution (molarity based) and a hypothetical pure H$^+$ solution (mole fraction based) are also often used when quantifying aerosol acidity. Supplement Table S1 provides a brief summary of studies reporting aerosol pH calculated using thermodynamic models with different definition of pH. Molality based pH, as suggested by IUPAC, is used in 12 out of 32 studies. Molarity-based pH is the most commonly used scale in aquatic chemistry as the equilibrium constant is often determined based on molarity (Stumm and Morgan, 1996); it is also widely used for characterizing aerosol acidity (7 out of 32 studies). Mole fraction-based pH has also been used to characterize the acidity of hygroscopic aerosols (5 out of 32 studies) as this approach is more convenient to describe solutions with high concentrations (Rard et al., 2010).

It appears that the selection of the standard state of activity is arbitrary for aerosol acidity studies, and is not always defined in published articles when pH is used to characterize the acidity of aerosol (8 out of 32 studies as shown in Table S1). This may not be problematic in the case of ISORROPIA-II where the default output pH is always molality-based; however, confusion is possible when E-AIM or AIOMFAC are used as these models provide both molality- and mole fraction-based concentrations as output. In fact, pH based on different definitions have sometimes been used in the same study; e.g., Hennigan et al. (2015) defined pH based on the mole fraction of hydrogen; however, the authors used pH=7 as the critical point when [H$^+$] = [OH$^-$], which actually is an elaboration of molarity (or molality) based pH. Some studies have employed molarity and molality of H$^+$ interchangeably in terms of defining and calculating pH (defined as mol dm$^{-3}$ of H$^+$ but calculated as mol kg$^{-1}$ of H$^+$, e.g., Guo et al., 2016), which is not ideal for the sake of consistency even though the resultant estimates are comparable. Additionally, pH values obtained via different definitions are sometimes cross-compared, e.g., Squizzato et al. (2013) observed that pH of PM$_{2.5}$ in the Po Valley, Italy (mole fraction-based) was much higher than those in megacities in China (Pathak et al., 2009) (molarity-based). Such comparisons need to be reevaluated given the different definitions of pH adopted in these studies.

Despite apparent incongruities in such cross-comparisons, this issue has not been addressed with sufficient care by the atmospheric science community. The main objective of this study is thus to compare PM$_{2.5}$ aerosol pH based on different scales (molarity, molality and mole fraction) on the basis of theoretical considerations followed with a set of field data as an example. Further, in order to enable other researchers to easily compare pH based on different scales, the use of an inter-scale conversion factor has been demonstrated for the three most commonly used thermodynamic models, i.e., E-AIM-IV, ISORROPIA-II and AIOMFAC.

## 2 Materials and methods

### 2.1 Evaluation data set

A set of field data collected in Guangzhou, China was used to demonstrate the interconversion of pH based on different scales. The sampling site was located at the rooftop of a building, 15 m above the ground, in the Guangzhou Environmental Monitoring Center (23°07′59″ N, 113°15′35″ E) (refer to Chen et al., 2016b for details). Hourly ionic species of PM$_{2.5}$ were measured using an AIM-IC 9000D (URG, Chapel Hill, NC) (refer to Chen et al., 2016a for details). The sampling duration was from 1–31 July 2013.

## 2.2 Thermodynamic models

The three most widely used thermodynamic models including E-AIM-IV (http://www.aim.env.uea.ac.uk/aim/aim.php TS5) (Friese and Ebel, 2010; Wexler and Clegg, 2002), ISORROPIA-II (http://isorropia.eas.gatech.edu/index.php TS6) (Fountoukis and Nenes, 2007) and AIOMFAC (http://www.aiomfac.caltech.edu TS7) (Zuend et al., 2008) were selected to demonstrate the interconversion of pH between different scales. E-AIM is usually considered to be a benchmark model (Seinfeld and Pandis, 2016), while ISORROPIA is preferred for use in large-scale atmospheric models as it employs various simplifications to enhance computational efficiency (Fountoukis and Nenes, 2007). AIOMFAC can be used to calculate inorganic–organic interaction (Pye et al., 2018).

E-AIM-IV and ISORROPIA-II were run in forward mode (closed system). The compounds included in the calculation were $Cl^-$, $SO_4^{2-}$, $NO_3^-$, $NH_4^+$ and $Na^+$ in the particulate phase and $NH_3$, $HNO_3$ and $HCl$ in the gaseous phase. Other inorganic ions such as $K^+$, $Ca^{2+}$ and $Mg^{2+}$, and organic compounds were not included in the calculation in order to keep the consistency of the three models as $K^+$, $Ca^{2+}$ and $Mg^{2+}$ are not included in the system of E-AIM-IV while organic compounds are not included in ISORROPIA-II. This might induce some uncertainty in the estimated pH; however, this is not further discussed as the method to calculate aerosol acidity is not the focus of current study. The current online version of AIOMFAC is not capable of calculating gas-aerosol equilibrium, and thus the output of aerosol compounds from E-AIM-IV were used as input in AIOMFAC to obtain aerosol properties in the reverse mode (open system). A stable particle phase state (solid plus liquid) was assumed for E-AIM-IV and ISORROPIA-II. Compounds in the aqueous phase of the output of E-AIM-IV were used as input to AIOMFAC. That way, AIOMFAC can be considered to be consistent with E-AIM-IV and ISORROPIA-II. According to Song et al. (2018), SORROPIA-II calculations with resultant pH of close to neutral (in stable mode) may not be accurate; hence, these samples (303 out of 440) were excluded from the calculation for all three models.

## 2.3 pH calculation and interconversion

We provide below parameterizations of pH based on different standard states (molar fraction, molarity and molality). The reference state for the activity coefficients of $H^+$ ion is the infinite dilute solution in a reference solvent. Abbreviations used in this study are summarized in the Appendix.

$$pH_x = -\log_{10}(a_{x_H}) = -\log_{10}(f_H x_H) \tag{1}$$

$$pH_c = -\log_{10}(a_{c_H}) = -\log_{10}\left(\frac{y_H c_H}{c^o}\right) \tag{2}$$

$$pH_m = -\log_{10}(a_{m_H}) = -\log_{10}\left(\frac{\gamma_H m_H}{m^o}\right) \tag{3}$$

The equations for interconversion of $H^+$ concentrations and corresponding activity coefficients based on different standard states are listed in Table 1.

A number of parameters needed to estimate aerosol pH cannot be obtained directly from the three models, and calculations and/or assumptions are thus necessary. The details of the approach to obtain specific parameters are shown in Table S2, and pH of different scales are calculated based on their definitions (Eqs. 1–3). It is worthwhile to note that the molality based activity coefficient of $H^+$ in ISORROPIA-II is assumed to be 1; consequently, the activity coefficient of $H^+$ based on molarity and mole-fraction scale cannot be obtained and was also assumed to be 1. Moreover, the density of aerosol is not calculated by ISORROPIA-II or AIOMFAC, and thus the density output by E-AIM-IV were used for all the three models.

## 3 Results and discussion

### 3.1 Comparison of $pH_x$, $pH_c$ and $pH_m$

#### 3.1.1 Comparison of pH calculated by different models

The results of pH calculated based on different standard states with the three thermodynamic models are shown in Table 2. Overall, there are slight differences between pH calculated using different models. Taking $pH_m$ as an example, the averaged $pH_m$ calculated by ISORROPIA-II ($2.77\pm0.36$) is 0.25 pH unit higher than that calculated by E-AIM-IV ($2.52\pm0.28$), which is consistent with the result reported by Song et al. (2018) and Liu et al. (2017). The $pH_m$ calculated by AIOMFAC ($2.56\pm0.27$) is closer to that calculated with E-AIM-IV ($2.52\pm0.28$). It is worthwhile to note that the activity coefficient of $H^+$ calculated by E-AIM-IV ($0.57\pm0.19$) is 2.7 times higher than that calculated by AIOMFAC ($0.21\pm0.08$) while the molality of $H^+$ calculated using AIOMFAC (($1.98\pm2.50)\times10^{-2}$) is 2.5 times higher than that calculated by E-AIM-IV (($7.80\pm9.52)\times10^{-3}$) although the resultant $pH_m$ is similar.

The difference in the calculated pH between different models is due to differences in both concentration and activity coefficient. For example, a unity activity coefficient of $H^+$ is assumed for ISORROPIA-II for pH calculation even though the non-ideal interaction of $H^+$ with all other ions in solution is explicitly considered by the Kusik-Meisner and Bromely formulations in ISORROPIA-II (Fountoukis and Nenes, 2007). The exact factors contributing to the difference in pH remains unclear, and is not the focus of current study. The models may differ in many ways such as their methods for calculating the activity coefficients for $H^+$ and other ionic species, and in estimating aerosol water contents (Song et al., 2018).

**Table 1.** Summary of equations for the interconversion of concentration and activity coefficient based on different standard states. TS8

| Parameter | pH$_x$ vs. pH$_m$ | pH$_m$ vs. pH$_c$ | pH$_x$ vs. pH$_c$ |
|---|---|---|---|
| Activity coefficient[a] | $\gamma_H = f_H \frac{x_H}{m_H M_s}$ (4) | $\gamma_H = 1000 \frac{dm^3}{m^3} \frac{c_H y_H}{m_H \rho_0}$ (5) | $f_H = y_H 1000 \frac{dm^3}{m^3} \frac{M_s}{\rho_0} \frac{c_H}{x_H}$ (6) |
| Concentration[b] | $x_H = \frac{m_H}{\sum m_i + \frac{1}{M_s}}$ (7) | $c_H = \frac{m_H}{\frac{\sum m_i M_i + 1}{\rho_{sln}}}$ (8) | $x_H = \frac{M_s c_H}{M_s \sum c_i + 0.001 \frac{m^3}{dm^3} \rho_{sln} - \sum c_i M_i}$ (9) |
| pH[c] | $pH_x - pH_m = -\log_{10}[M_s m^0$ TS9$]$ (10) | $pH_m - pH_c = -\log_{10} \frac{c^0 1000\, dm^3/m^3}{m^0 \rho_0}$ (11) | $pH_x - pH_c = \log_{10} \frac{1000\, dm^3/m^3 M_s c^0}{\rho_0}$ (12) |

Note: [a] The source of Eqs. (4)–(5) are Robinson and Stokes (2002) and the source of Eq. (6) is Zuend (2007). The details of derivation of Eqs. (4)–(6) are shown in Robinson and Stokes (2002) and Zuend (2007). [b] Equations (7)–(9) are based on the definition of each parameter. [c] Equations (10)–(12) are derived from Eqs. (4)–(6) and (7)–(9) for each standard state.

**Table 2.** Calculated concentrations, activity coefficient of H$^+$ and pH for the three thermodynamic models.[a]

| | E-AIM-IV | ISORROPIA-II | AIOMFAC |
|---|---|---|---|
| **Molar fraction** | | | |
| $x_H$ | $(1.07 \pm 1.28) \times 10^{-4}$ | $(3.49 \pm 4.80) \times 10^{-5}$ | $(2.17 \times 10^{-5}$–$9.49 \times 10^{-4})$ |
| | $(2.71 \pm 3.36) \times 10^{-4}$ | $(4.59 \times 10^{-6}$–$3.69 \times 10^{-4})$ | $(4.56 \times 10^{-5}$–$2.46 \times 10^{-3})$ |
| $f_H$ | $0.74 \pm 0.25$ | | $0.27 \pm 0.10$ |
| | $(0.43$–$1.97)$ | 1[b] | $(0.15$–$0.79)$ |
| $pH_x$ | $4.26 \pm 0.28$ | $4.63 \pm 0.36$ | $4.31 \pm 0.27$ |
| | $(3.16$–$4.82)$ | $(3.43$–$5.34)$ | $(3.24$–$4.86)$ |
| **Molality** | | | |
| $m_H$ | $(7.80 \pm 9.52) \times 10^{-3}$ | $(2.60 \pm 3.64) \times 10^{-3}$ | $(1.98 \pm 2.50) \times 10^{-2}$ |
| | $(1.50 \times 10^{-3}$–$7.03 \times 10^{-2})$ | $(3.18 \times 10^{-4}$–$2.80 \times 10^{-2})$ | $(3.14 \times 10^{-3}$–$1.82 \times 10^{-1})$ |
| $\gamma_H$ | $0.57 \pm 0.19$ | | $0.21 \pm 0.08$ |
| | $(0.35$–$1.54)$ | 1[b] | $(0.12$–$0.62)$ |
| $pH_m$ | $2.52 \pm 0.28$ | $2.77 \pm 0.36$ | $2.56 \pm 0.27$ |
| | $(1.41$–$3.07)$ | $(1.55$–$3.50)$ | $(1.50$–$3.11)$ |
| **Molarity** | | | |
| $c_H$ | $(5.56 \pm 6.59) \times 10^{-3}$ | $(1.73 \pm 2.35) \times 10^{-3}$ | $(1.43 \pm 1.76) \times 10^{-2}$ |
| | $(1.14 \times 10^{-3}$–$4.89 \times 10^{-2})$ | $(2.38 \times 10^{-4}$–$1.80 \times 10^{-2})$ | $(2.48 \times 10^{-3}$–$1.30 \times 10^{-1})$ |
| $y_H$ | $0.79 \pm 0.26$ | | $0.28 \pm 0.10$ |
| | $(0.45$–$2.04)$ | 1[b] | $(0.16$–$0.75)$ |
| $pH_c$ | $2.52 \pm 0.28$ | $2.94 \pm 0.35$ | $2.56 \pm 0.27$ |
| | $(1.41$–$3.07)$ | $(1.75$–$3.62)$ | $(1.50$–$3.11)$ |

Note: [a] All parameters are shown as average ± standard deviation with the range in bracket except for [b] activity coefficient of ISORROPIA-II which is assumed to be 1.

### 3.1.2 Comparison of pH based on different scales

As ISORROPIA-II simplifies the calculation with the assumption of the activity coefficient as unity while E-AIM and AIOMFAC calculate the activity coefficients in practice, ISORROPIA-II is discussed separately from the other two models in the following text.

For E-AIM-IV and AIOMFAC, the interconversion of pH based on different standard states can be conducted based on the theory (Eqs. 10–12) (e.g., Robinson and Stokes, 2002) as all parameters are available. The difference of pH$_x$ and pH$_m$ is $\log_{10} M_s m^0$ (according to Eq. 10), which is only determined by the molecular weight of the solvent. When water is the only solvent in the system (molecular weight of 0.018 kg mol$^{-1}$), pH$_x$ − pH$_m$ is fixed at 1.74 within the model regardless of aerosol property or the model (as in this study). As shown in Table S1, water is taken as the only solvent in aerosol solution in almost all studies. The only study that considers organic compounds as one of the solvent is Pye et al. (2018).

In contrast, the difference between pH$_c$ and pH$_m$, $\log_{10}(1000 \frac{dm^3}{m^3} \frac{c^0}{m^0 \rho_0})$, is related to the density of the pure solvent (Eq. 11), while the difference between pH$_x$ and pH$_c$, $\log_{10} \frac{1000\, dm^3/m^3 M_s c^0}{\rho_0}$, is determined by both the molecular weight and the density of the pure solvent (Eq. 11). As standard states are defined at the same temperature and pres-

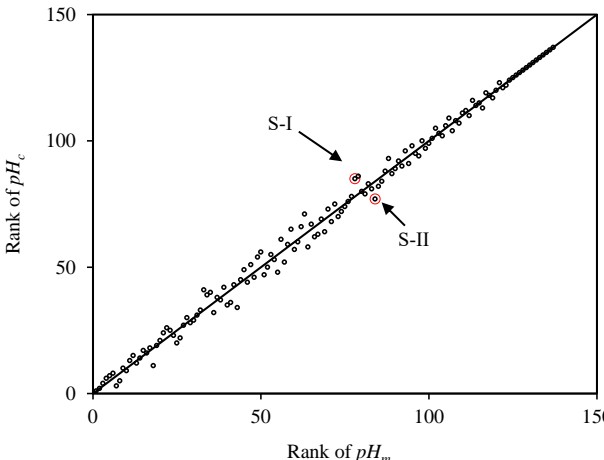

**Figure 1.** Comparison of the rank of $pH_m$ and $pH_c$.

**Table 3.** Comparison of acidity of selected samples based on different scales.

| # | $pH_m$ | $m_H$ | $pH_c$ | $c_H$ |
|---|---|---|---|---|
| S-I | 2.70 | $2.01 \times 10^{-3}$ | 2.92 | $1.21 \times 10^{-3}$ |
| S-II | 2.75 | $1.80 \times 10^{-3}$ | 2.87 | $1.34 \times 10^{-3}$ |
| Difference ($\Delta$) | $-0.05$ | $2.0 \times 10^{-4}$ | 0.05 | $-1.3 \times 10^{-4}$ |

sure as the solution (Robinson and Stokes, 2002), the density of a pure solvent can vary at standard state for different solutions based on corresponding temperature and pressure. However, the density of water (the major solvent in atmospheric aerosols) does not vary significantly with temperature and pressure. The variation of water density is only 4 % within a temperature ranging from 0 to 100 °C (Kell, 1975) (corresponding to a change of pH is only 0.02 pH unit). The change of water density due to pressure variation is even smaller. When pressure ranges from 0.1 to 10 MPa at 25 °C, the density change is only 0.004 % (Wagner and Pruß, 2002) (corresponding pH change is $1.9 \times 10^{-4}$). Therefore, the difference can be neglected for water at ambient temperature and pressure. While the temperature ranges from 24.55 to 31.55 °C in the current study, the water density varies from 9.952 to $9.972 \times 10^4$ Pa, with the corresponding pH change being less than 0.001 pH unit.

However, for ISORROPIA-II, the activity coefficient is assumed to be unity for the molality scale. If the same assumption is made for the other scales, the conversion factor deviates somewhat from theory. As shown in Table 2, the averaged $pH_m$ (2.77) is 0.15 pH unit (ranging from 0.11 to 0.25) lower than $pH_c$ (2.94) due to the simplification of both activity coefficients as unity. This effect is of a similar magnitude to that of organic-associated water to aerosol pH (0.15 to 0.23 pH unit) (Guo et al., 2015). Based on Eq. (8), the difference between $pH_m$ and $pH_c$ is not only related to the concentration of other species, but is also affected by the density of the solution (Eq. 8). The density of the solution in turn varies with relative humidity and chemical properties of the samples (Clegg and Wexler, 2011), thus leading to potential variations in the trend of $pH_m$ and $pH_c$. To investigate the trend comparison between $pH_m$ and $pH_c$, their ranks (in descending order) are plotted in Fig. 1. The points deviating from the 1 : 1 line indicate samples possessing different ranks according to $pH_m$ compared to that of $pH_c$. To illustrate how pH

trends could change with different scales, two samples which deviate most from the 1 : 1 line are selected as examples (marked S-I and S-II in Fig. 1). As shown in Table 3, S-I is more acidic than S-II upon comparison of $pH_m$ values. However, in terms of $pH_c$, S-I is less acidic than S-II. Although $\Delta pH_m(-0.05)$ is only 0.1 pH unit lower than $\Delta pH_c(0.05)$, the difference in H$^+$ concentration may not be neglected. The molality of H$^+$ ions of S-I ($2.01 \times 10^{-3}$ mol kg$^{-1}$ water) is 11.7 % higher than that of S-II ($1.80 \times 10^{-3}$ mol kg$^{-1}$ water); however, the molarity of S-I ($1.21 \times 10^{-3}$ mol dm$^{-3}$ solution) is 10.7 % lower than that of S-II ($1.34 \times 10^{-3}$ mol dm$^{-3}$ solution). Given that the uncertainty of pH calculation due to measurement errors can be as high as 14 % (Guo et al., 2015), the difference of $pH_c$ and $pH_m$ can simply fall within the range of measurement errors. However, the bias between $pH_c$ and $pH_m$ can be considered to be systematic, which needs to be addressed for the sake of comprehensiveness in theoretical analysis. Moreover, even small biases in pH may imply substantial partitioning errors for semivolatile species like ammonium, nitrate, chloride and even organic acids (Guo et al., 2017). Therefore, while evaluating aerosol acidity (especially, trend analysis) when the activity coefficient is simplified as 1, considering the pH scale is important. For the conversion between $pH_x$ and $pH_m$, when the solvent is fixed as water, the difference is affected by the molality of H$^+$ and other electrolyte species in liquid water (according to Eq. 7). In the current study, the $pH_x$–$pH_m$ ranges from 1.83 to 1.87 which is 0.09 to 0.13 pH units higher than that based on theory (1.74). The trends of $pH_x$ and $pH_m$ can also be different but with a smaller difference compared with that between $pH_x$ and $pH_m$ as shown in Fig. S1 in the Supplement.

## 3.2 General issues with pH comparison

It has been shown above that proper scale conversion has to be conducted when aerosol pH is compared. However, one should bear in mind that even with the same measured data and scale, pH calculated with different thermodynamic models or with different parameters may still not be comparable. Below, we briefly describe some of the general issues that need to be considered when aerosol acidity is compared across studies along with a summary of parameters used in the published studies in Table S1.

Thermodynamic models like ISORROPIA-II and E-AIM can run in forward mode and reverse mode that results in

significant differences (Song et al., 2018; Hennigan et al., 2015). It is recommended to use thermodynamic models in forward mode (gas plus aerosol as input) instead of reverse mode (aerosol only as input), which is highly sensitive to measurement uncertainties (Hennigan et al., 2015).

Thermodynamic models can also be run in stable (liquid only) or metastable modes (both solid and liquid), which has not been specified in many studies (Table S1). Song et al. (2018) have shown that model calculations coupled with stable or metastable state assumptions can provide reasonable estimates of aerosol water and pH. However, as pointed by Song et al. (2018), the studies using standard ISORROPIA-II (without code correction) running in stable mode have predicted unrealistic pH values of around 7 and should be reevaluated.

The effect of non-volatile cations such as $Na^+$, $Ca^{2+}$, $Mg^{2+}$ and $K^+$ on aerosol pH may also not be ignored. Although the effect of non-volatile cations on pH may be only 0.2–0.5 pH units, their impact on predicted partitioning of a semi-volatile species can be significant due to the highly non-linear response of $NH_3$-$NH_4^+$ or $HNO_3$-$NO_3^-$ partitioning to pH (Guo et al., 2017). As E-AIM cannot explicitly treat $Ca^{2+}$, $Mg^{2+}$ and $K^+$ (unlike ISORROPIA-II and AIOM-FAC), pH estimated using E-AIM may ignore $Ca^{2+}$, $Mg^{2+}$ and $K^+$ (as shown in Table S1) or treat them as equivalent sodium (e.g., Hennigan et al., 2015). Even if all non-volatile cations are treated as $Na^+$, the predicted thermodynamic states can be biased due to the strong non-ideality of divalent ions as well as variations in water uptake characteristics between $Na^+$ salts and its counterparts (Fountoukis et al., 2009).

Most studies so far have estimated pH of aerosols with only inorganic compounds. However, it has been reported that pH can be affected by organic compounds in several ways. Guo et al. (2015) have shown that the pH can be increased by 0.15 to 0.23 units when aerosol water associated with organic compounds is considered. Omission of the contribution of organic acids to $H^+$ has been estimated to increase the pH by $0.07 \pm 0.03$ by Song et al. (2018) using E-AIM-IV. It has been shown recently that accounting for non-ideal mixing can modify the pH such that a fully interactive inorganic–organic system showed a pH roughly 0.7 units higher than that predicted using an inorganic only system by AIOMFAC (Pye et al., 2018).

## 4   Conclusions

This study compares aerosol pH based on three different standard states ($pH_x$, $pH_m$ and $pH_c$) and the corresponding interconversion. Established theory dictates that the difference between $pH_x$ (mole fraction based) and $pH_m$ (molality based) is always a constant within a thermodynamic model (1.74, when the solvent is water) regardless of aerosol property. In contrast, $pH_m$ and $pH_c$ (molarity based) are almost

identical with a minor effect from temperature and pressure. However, when the activity coefficient is simplified as unity by thermodynamic models, the difference between $pH_m$ and $pH_c$ ranges from 0.11 to 0.25 pH units, depending on the chemical composition and density of hygroscopic aerosol. Therefore, while evaluating aerosol acidity (especially, trend analysis) when the activity coefficient is simplified as 1, considering the pH scale is important. Overall, we recommend that the standard state of hydrogen activity be defined clearly when pH values are used to characterize the acidity of aerosol, and that pH values are converted to the same scale prior to comparison of acidity. As suggested by Nenes (2018), maintaining consistency in terms of pH calculation method and the thermodynamic model used is vital to ensure comparability of aerosol acidity between models and/or observations.

*Data availability.* Data for this paper are available from the corresponding authors upon request.

## Appendix A: List of abbreviations.

| Abbreviation | Definition |
| --- | --- |
| $ac_H$ | activity of hydrogen ions with standard state of the hypothetical ideal aqueous solution of unit molarity and reference state of infinite dilute solution (dimensionless) |
| $am_H$ | activity of hydrogen ions with standard state of the hypothetical ideal aqueous solution of unit molality and reference state of infinite dilute solution (dimensionless) |
| $ax_H$ | activity of hydrogen ions with standard state of the hypothetical pure $H^+$ solution and reference state of infinite dilute solution (dimensionless) |
| $c^0$ | unit molarity (1 mol dm$^{-3}$ solution) |
| $c_H$ | molarity of hydrogen ion (mol dm$^{-3}$ solution) |
| $c_i$ | molarity of solute species $i$[a] (mol dm$^{-3}$ solution) |
| $f_H$ | mole fraction scale activity coefficient |
| $m^0$ | unit molality (1 mol kg$^{-1}$ solvent) |
| $m_H$ | molality of hydrogen ions (mol kg$^{-1}$ solvent) |
| $m_i$ | molality of solute species $i$[a] (mol kg$^{-1}$ solvent) |
| $M_i$ | molar mass of solute species $i$[a] (kg mol$^{-1}$) |
| $M_s$ | molar mass of single solvent or averaged molar mass for multiple solvents (kg mol$^{-1}$) |
| $pH_c$ | molarity based pH (dimensionless) |
| $pH_m$ | molality based pH (dimensionless) |
| $pH_x$ | mole fraction based pH (dimensionless) |
| $x_H$ | mole fraction of hydrogen ions (dimensionless, mol $H^+$ in total moles) |
| $y_H$ | molarity scale activity coefficient |
| $\gamma_H$ | molality scale activity coefficient |
| $\rho_0$ | density of pure solvent or averaged density for multiple solvents (kg m$^{-3}$) |
| $\rho_{sln}$ | density of hygroscopic aerosol solution (kg m$^{-3}$) |

[a] Solute species $i$ is expressed as dissociated ion for salt.

**The Supplement related to this article is available online at https://doi.org/10.5194/acp-18-1-2018-supplement.**

*Author contributions.* TS10

*Competing interests.* The authors declare that they have no conflict of interest. TS11

*Acknowledgements.* The authors thank Krishnan Padmaja for her helpful comments and discussion during the preparation of the manuscript. This work has been funded by the (1) National Key Research and Development Program of China under 2017YFC0210105, (2) National Science Fund for Distinguished Young Scholars under 41425020, (3) Science and Technology Planning Project of Guangdong Province, China under 2016B050502005 and (4) Guangdong Natural Science Foundation under 2017A030313234.

Edited by: Rob MacKenzie
Reviewed by: Andreas Zuend and one anonymous referee

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

Please note the remarks at the end of the manuscript.

**Remarks from the typesetter**