# Peer review of "Technical note: Comparison and interconversion of pH based on different standard states for aerosol acidity characterization"

_Atmospheric Chemistry and Physics, 2018_

## Short Comment (SC1) · 2 Feb 2018

I believe I reviewed a very similar ms to this one (submitted to another journal) in 2017. I presume it was rejected.

Below are what I see as some of the most important issues regarding our understanding (from measurement or from models) of aerosol acidity. I don't think that the authors really grasp them or the problems involved, and that this proposed Technical Note does not provide the clarity and guidance that are needed by the field.

1. Any discussion of pH ought to start with a recognition of what it is, which is not some

"absolute" but depends on either how it is measured or the model used to calculate it. See for example the first sentence of Buck et al. (Pure Appl. Chem. 74, 2169-2200, 2002):

"The concept of pH is unique among the commonly encountered physicochemical quantities listed in the IUPAC Green Book in that, in terms of its definition,

pH = -lg a

it involves a single ion quantity, the activity of the hydrogen ion, which is immeasurable by any thermodynamically valid method and requires a convention for its evaluation."

So, for example, values of model-calculated H+ activities depend on how the model "splits" mean activity coefficients of cation-anion pairs (which are measurable) into single ion activities (which are not). This many not be a problem if all the calculations of thermodynamic properties (gas/aerosol equilibrium for example) are made with one model, but it should nonetheless be recognised. The measurement of pH is tied to the defined pH of the buffer used to calibrate the instrument. The buffer needs to be chemically similar to the solution being measured. Measured and modelled pH may not be comparable.

2. The H+ content of an aerosol should never be calculated from charge balance, for reasons that are so obvious that they hardly need repeating here (chemical analysis may not include all charged species; real H+ content is likely to be small relative to the amounts of the other cations and anions and is obtained as the [small] difference between two large and uncertain numbers). The use of measurements of gas phase NH3 or HNO3, in addition to the major ion composition of the aerosol, as constraints on aerosol acidity seems the sensible approach to me. I'm aware that the group of Jose Jimenez uses this method. I expect there are others.

3. The relationship between solute activities on different concentration scales - the basis of equations 1 to 10 of the authors - is textbook material. See for example

Chapter 2 of "Electrolyte Solutions" by Robinson and Stokes. The authors haven't cited this or any other chemistry textbook.

4. The rough correlation between pH on the molarity scale and on the other scales is to be expected, as is clear from eq (9), and this part of their discussion doesn't really add to the content of the ms.

5. Discussions of aerosol acidity need to address the fact that "pH" will go up and down with the diurnal cycle of RH, while the H+ content of the aerosol might remain more or less the same.

---

## Referee Comment (RC1) · A. Zuend (Referee) · 8 Mar 2018

**1   General comments**

Acidity is an important characteristic of liquid aerosol particle phases, which often tend to be highly acidic, as expressed by a low pH value. This technical note addresses several issues arising from the existence of distinct ways to define "pH". The authors discuss the differences between pH defined on molality, molarity and mole fractions scales and the importance of using thermodynamically correct conversions among these scales when field data is compared. This scale-dependence of pH is an important point indeed. While this reviewer has been aware of the pH scale dependence and its potential pitfalls for a while, it is an issue appropriate for a "technical note" outlining the proper thermodynamic scale conversion (theory) and providing discussion of related practical issues in aerosol acidity evaluations from field data.

However, the present manuscript contains a number of flaws, several of which are further discussed under specific comments below. The interactive comment by Simon Clegg (https://www.atmos-chem-phys-discuss.net/acp-2018-85/acp-2018-85-SC1.pdf) summarizes the main concerns shared by this reviewer. Major revisions are necessary to transform this manuscript into a paper that contains (i) a thorough discussion of the thermodynamic scale conversions as well as (ii) an appropriate discussion of the general issues with pH estimation of field aerosol samples, given that the actual $H^+$ concentration is not routinely measured and other properties like water content and cation/anion balances may suffer from substantial measurement uncertainty and artifacts. While the latter point is not the focus of this technical note, mostly ignoring the issues of that point is not appropriate either. Errors from incorrect $H^+$ concentration estimations, e.g., by use of an ion charge balance, as done in this study (and others), may frequently be more substantial than the errors from pH value comparisons without proper pH scale conversion. Therefore, a discussion of issues with aerosol acidity determination and pH scale intercomparison must include both.

**2  Specific comments**

1. Abstract, line 16: Stating that this issue is addessed "for the first time" is rather bold given that the theoretical framework for activity coefficient and pH scale conversions has been known for decades (even though it may be true that it is a frequently ignored issue, therefore it is worthy of attention by the atmospheric chemistry community).

2. Abstract, line 18: "Using hourly ionic species measurements in Guangzhou, China, it is observed that $pH_x$ (mole fraction based) is always 1.74 pH unit higher than $pH_m$ (molality based)". This is clearly misleading, since the pH unit difference is not truly based on observation. It is a circular argument also made at other places in the manuscript. The 1.74 pH unit difference is in fact coming from the application of such a theory-based difference and using the same thermodynamic model to compute the pH values in different scales based on $H^+$ activity coefficient prediction from the same samples.

3. Page 2, line 9: The authors state; "The acidity of aerosols can be quantified by parameters such as strong acidity, free acidity, cation-to-anion ratio and ammonium-to-sulfate ratio. However, these parameters neglect the effect of liquid water content or the dissociation of ions and acids (Pathak et al., 2004; Hennigan et al., 2015)." First, a definition of what strong acidity and free acidity represents is not given. Second, it is incorrect that free acidity neglects water content, as the partial dissociation of species like bisulfate is very much dependent on water content and therefore affects free acidity. Pathak et al. and Hennigan et al. do not seem to make such a point.

4. Equation (1): Define the meaning of "lg".

5. Equation (2): This equation and its description is flawed. (1) What are the terms of "$1000$" in the numerator and denominator? The authors likely use these for conversion from units of kg to g. If so, the mathematically correct way of writing this would be to write $1000 \frac{g}{kg}$ in the equation and it would be necessary to state that the molar masses are supposed to be used in units of $\frac{g}{mol}$ rather than the standard SI unit of $\frac{kg}{mol}$. Otherwise it is simply incorrect and a potential source of confusion. However, since the ACP recommendation is to use SI units whenever possible, there is not need for these unit conversion terms at all (they would be 1). (2) This expression is only correct for the special case where the only

solvent for the ions is water. However, in the more general case, there may be other solvents, such as organic compounds mixed with water and the ions in a liquid phase. In that case, the distinct molar masses of the organics must be accounted for in the activity coefficient conversion expression. Hence, since this is a key part of the discussion about different thermodynamic composition scales and reference and standard states, it should be shown correctly for the general case. A rigorous derivation of such scale conversions is, e.g., shown in the PhD thesis by Zuend (2007) (page 45 – 47 there), which shows different versions of the scale conversion formula. One of which (useful here) is $\ln[\gamma_i] = \ln[f_i^*] + \ln[\frac{x_i}{m_i M_w}]$ where $\gamma_i$ is the molality scale activity coefficient of ion "i" (e.g. $H^+$) and $f_i^*$ the mole fraction scale activity coefficient, both with reference state of infinite dilution in pure water; $x_i$ the ion mole fraction with respect to dissociated ions, $m_i$ the ion molality and $M_w$ the molar mass of water. Since solvents other than water are also included in both the mole fraction and molality expressions for ion "i", this is a general expression. From this scale conversion of ion activity coefficients, it is readily shown that the difference in pH values is generally given by $pH_x - pH_m = -\log_{10}[m^\circ M_w]$, where $m^\circ$ denotes unit molality (= 1 mol/kg) (similar to Eq. (8) in the manuscript, but note the difference in units, the given Eq. (8) is not dimensionless in the log).

6. As pointed out in the comment by Simon Clegg, thermodynamic models differ in the way single-ion activity coefficients are calculated (since only mean cation/anion pair activity coefficients are measureable) and of course they also differ in the expressions, such that even with the correct conversion of activity coefficients or pH values between scales, different models may predict different $pH_m$ (or $pH_x$) values for the same input composition. Furthermore, only some models account for the influence of organic species in the mixture (e.g. AIOMFAC can be used for that, while ISORROPIA is only for inorganic aqueous mixtures) and differences in predicted pH may partially stem from organic interactions with ions

and treatment of phase separation, see Pye et al (2018).

7. Equation (4): Similar to above comment. A general expression should be shown, with proper use of units.

8. Page 4, line 3: the last sentence there makes little sense. The pH values can be compared when the scale effect is accounted for; the point is that one should not expect the values to be equal.

9. Page 4, line 13: "This is supported by our field data". Again, this is a circular argument. The $pH_x - pH_m$ scale difference is used in the evaluation of the pH values, so of course it will show as consistent, but the measurements are no proof for that. Also, the fact that the difference should be a constant in pH units is clear from the theory, as long as the same thermodynamic model is used to compute the activity coefficients (which may not be the case when different studies are compared).

10. Page 7, line 6: The discussion in this paragraph is not sufficient to address the other very important issues when aerosol acidity is attempted to be assessed from field measurements. It is also clear from theory and comparisons that organic compounds will affect aerosol acidity, maybe not dramatically but noticeably, since their interactions with water and $H^+$ ions are affecting the activity of $H^+$. Last sentence in paragraph: "The relationship between pHx, pHc and pHm established in this study is valid regardless of the method selected to estimate aerosol acidity." This is true only for the scale conversion, since it depends on proper application of thermodynamic theory only (which has been known for decades and is not a novelty of this study). However, when aerosol sample acidity is estimated in practice, different models are used for activity coefficients (e.g. ISORROPIA, E-AIM, AIOMFAC) or unit activity coefficients are assumed (not recommended). Moreover, different methods are applied to determine the approximate $H^+$ amount, which is a critical problem in acidity evaluation, see Hennigan

et al (2015). Therefore, the difference in reported pH values is not just due to the offsets between these different pH scales. The authors have failed to make this important point very clear.

11. Table 1: The definitions include many mistakes and typos; e.g., pH$_m$ is not molarity based, the last two entries are confusing and not correctly described and reference states of activity coefficients are missing.

**References**

- Hennigan, C. J., Izumi, J., Sullivan, A. P., Weber, R. J., and Nenes, A.: A critical evaluation of proxy methods used to estimate the acidity of atmospheric particles, Atmos. Chem. Phys., 15, 2775-2790, 10.5194/acp-15-2775-2015, 2015.

- Pathak, R. K., Yao, X., and Chan, C. K.: Sampling Artifacts of Acidity and Ionic Species in PM2.5, Environ. Sci. Technol., 38, 254-259, 10.1021/es0342244, 2004.

- Pye, H. O. T., Zuend, A., Fry, J. L., Isaacman-VanWertz, G., Capps, S. L., Appel, K. W., Foroutan, H., Xu, L., Ng, N. L., and Goldstein, A. H.: Coupling of organic and inorganic aerosol systems and the effect on gas–particle partitioning in the southeastern US, Atmos. Chem. Phys., 18, 357–370, doi:10.5194/acp-18-357-2018, 2018.

- Zuend, A.: Modelling the Thermodynamics of Mixed Organic-Inorganic Aerosols to Predict Water Activities and Phase Separations, Phd thesis, ETH Zurich, Zurich, Switzerland, doi:10.3929/ethz-a-005582922, http://e-collection.ethbib.ethz.ch/view/eth:30457, 2007.

---

## Referee Comment (RC2) · Anonymous Referee #2 · 20 Mar 2018

Particle acidity is an important property in terms of aerosol chemistry and its impact on health and climate. Due to the importance, there has been an increasing amount of publication investigating particle pH through thermodynamic models in recent years, which is a step forward compared to the problematic ion balance or molar ratio because of particle liquid water and non-ideality effect (activity coefficient). Without a consensus on the pH definition, it would be difficult to compare various studies, and this may lead to misunderstanding or misinterpretation of particle pH. Although some particle acidity studies state clearly the pH definition used in the analyses, there are some papers not talking about the definition at all. In the latter case, some studies probably take the approach of molality-based or molarity-based because they take a pH of 7 as neutral.

[Figure]

This paper shows the two pHs are very close and the minor difference is caused by particle density (for dilute water solution, no difference is expected from the two pHs). In general, this paper hits an important point (i.e., pH definition) that has not been paid enough attention to by the community and fits the scope of a technical note on ACP. However, some revisions are needed before considering a publication.

There are several "circular" statements as pointed out by the other reviewer Andreas Zuend. The differences between pHx, pHc, pHm are all expected from their definitions. This is the key point of this paper: the difference between pHc and pHm is small (within 0.2 pH units, caused by particle density), but pHx is significantly higher than pHc and pHm (pHx – pHm = 1.74). The E-AIM (or any other model) predicted pHs are supposed to be consistent with the rule, if one model is used consistently in this analysis. Therefore, it is not a real support by field data (used as E-AIM input) as claimed on line 13 Page 5. Relating to this, the author should consider changing the saying of "observed" (Line 18 Page 1) in the abstract. The presented result is all based on a thermodynamic model prediction of pH but not measured pH.

From a boarder view of an application, this paper could be more beneficial by showing the default pH scales given by widely used thermodynamic models. The E-AIM model has been discussed in detail, however, ISORROPIA or AIOMFAC or any other model is not mentioned. For example, ISORROPIA gives pH in the model output and the pH scale is molality-based (Fountoukis et al., 2009; Guo et al., 2015). If a literature doesn't specify the pH scale, this piece of info could be very helpful to readers.

The section 3.3 is problematic without discussion of uncertainty, especially considering that the presented pH in this study is solely based on particle data (no gas data used to constrain pH). The predicted pH uncertainty is propagated from the particle ionic composition data (6% reported by Chen et al. (2016)) and RH, T. It is not easy to estimate particle pH uncertainty. Guo et al. (2015) estimated a pH uncertainty of 13% using another model, ISORROPIA, and in forward model for their dataset. Even though the two data points (S-I and S-II) are selected for the largest deviation from the

1:1 line, small differences (7% or 8%) in hydrogen ion activity are found, which seem to be within uncertainty range. Hennigan et al. (2015) and other papers have pointed out that forward mode is superior over reverse mode in terms of particle pH prediction accuracy. For one reason, reverse mode is more sensitive to particle measurement uncertainty (likely the cause for occasionally very high pH seen in Figure 1). For the other reason, particle pH is sensitive to gas-particle partitioning of semivolatile species (e.g., $NH_4^+$, $NO_3^-$, and $Cl^-$), as long as the species is not totally in gas phase or particle phase. The forward mode predicted pH can then be validated if predicted and observed gas-particle partitioning agree. In contrast, the reverse mode predicted pH could not be verified in the same way. The Line 7 on Page 7 seems to suggest there is no advantage of using forward mode calculation and this is misleading to potential users.

As Andreas Zuend points out, it would be nice to mention the possible differences between predicted pH via different models at the end of section 3.4. Even if the same inputs are given to models, 100% agreement in pH is not expected due to differences in assumptions and approaches. There are a number of studies comparing thermodynamic models and exploring the differences (Hennigan et al., 2015; Liu et al., 2017; Pye et al., 2018; Song et al., 2018).

Minor comments:

- Toning down the statement of "for the first time" is suggested. Thermodynamic specialists must have known the difference between pH scales. The elucidation in the paper is based on established equations on textbooks and doesn't sound to me like a groundbreaking finding. The paper is of value to minimize the gaps in the current understanding of pH definitions.

- The units in Equation (3) (mol/kg water) and (6) (mol/dm3) look redundant. However, more explanations in the text are needed. Equation (3) is defined based on the mass of water, while Equation (6) is defined based on the volume of particle, which includes

the volumes of water (solvent) and other solutes. If Equation (6) is defined solely on the volume of particle water, there would not be any difference with Equation (3). This is not clear in the text.

- Proper reference as suggested by Simon Clegg.

References:

Chen, W., et al.: Chemical Composition of PM2.5 and its Impact on Visibility in Guangzhou, Southern China, Aerosol Air Qual. Res., 16, 2349-2361, doi: 10.4209/aaqr.2016.02.0059, 2016.

Fountoukis, C., et al.: Thermodynamic characterization of Mexico City aerosol during MILAGRO 2006, Atmos. Chem. Phys., 9, 2141-2156, doi: 10.5194/acp-9-2141-2009, 2009.

Guo, H., et al.: Fine-particle water and pH in the southeastern United States, Atmos. Chem. Phys., 15, 5211-5228, doi: 10.5194/acp-15-5211-2015, 2015.

Hennigan, C. J., Izumi, J., Sullivan, A. P., Weber, R. J., and Nenes, A.: A critical evaluation of proxy methods used to estimate the acidity of atmospheric particles, Atmos. Chem. Phys., 15, 2775-2790, doi: 10.5194/acp-15-2775-2015, 2015.

Liu, M., et al.: Fine particle pH during severe haze episodes in northern China, Geophys. Res. Lett., 44, 5213-5221, doi: 10.1002/2017gl073210, 2017.

Pye, H. O. T., et al.: Coupling of organic and inorganic aerosol systems and the effect on gas-particle partitioning in the southeastern US, Atmos. Chem. Phys., 18, 357-370, doi: 10.5194/acp-18-357-2018, 2018.

Song, S., Gao, M., Xu, W., Shao, J., Shi, G., Wang, S., Wang, Y., Sun, Y., and McElroy, M. B.: Fine particle pH for Beijing winter haze as inferred from different thermodynamic equilibrium models, Atmos. Chem. Phys. Disc., 1-26, doi: 10.5194/acp-2018-6, 2018.

---

## Author Comment (AC1) · 26 Jun 2018

**Response to Dr. Zuend**

*Comments are in black, responses in* *blue* *and the revised text in* *red*.

**1 General comments**

Acidity is an important characteristic of liquid aerosol particle phases, which often tend to be highly acidic, as expressed by a low pH value. This technical note addresses several issues arising from the existence of distinct ways to define "pH". The authors discuss the differences between pH defined on molality, molarity and mole fractions scales and the importance of using thermodynamically correct conversions among these scales when field data is compared. This scale-dependence of pH is an important point indeed. While this reviewer has been aware of the pH scale dependence and its potential pitfalls for a while, it is an issue appropriate for a "technical note" outlining the proper thermodynamic scale conversion (theory) and providing discussion of related practical issues in aerosol acidity evaluations from field data.

However, the present manuscript contains a number of flaws, several of which are further discussed under specific comments below.

The interactive comment by Simon Clegg (https://www.atmos-chem-phys-discuss.net/acp-2018-85/acp-2018-85-SC1.pdf) summarizes the main concerns shared by this reviewer. Major revisions are necessary to transform this manuscript into a paper that contains (i) a thorough discussion of the thermodynamic scale conversions as well as (ii) a discussion of the general issues with pH estimation of field aerosol samples.

While the latter point is not the focus of this technical note, mostly ignoring the issues of that point is not appropriate either. Errors from incorrect $H^+$ concentration estimations, e.g., by use of an ion charge balance, as done in this study (and others), may frequently be more substantial than the errors from pH value comparisons without proper pH scale conversion. Therefore, a discussion of issues with aerosol acidity determination and pH scale intercomparison must include both.

**Response:**

We thank Dr. Zuend for the constructive comments on our manuscript. We also thank him for agreeing with the importance of pH interconversion between different scales, which is the main focus of the current study.

First of all, we would like to apologize for the mistakes made in Eqs. 4 and 9 in the original manuscript that stem from errors originally made in the equation by van Boekel (2008) used in our reference. Since the mistakes are not included in either of the reviewer's comments or in the short comments by Prof. Clegg, we clarify this issue first before responding to the reviewers' comments in detail. The details of the mistakes are shown in Table 1 below.

Table 1. Correction of Eqs. 4 and 9 in the original manuscript

| Eq. # | Original (with mistake) | Corrected [a] |
|---|---|---|
| 4 | $y_H = f_H \dfrac{\rho_{\text{sln}}}{\rho_{\text{sln}} + 0.001[M_s\sum c_i - \sum c_i M_i]}$ | $y_H = f_H \dfrac{\rho_{solvent}}{\rho_{\text{sln}} + 0.001[M_s\sum c_i - \sum c_i M_i]}$ |
| 9 | $pH_x - pH_c = -\lg\dfrac{f_H x_H}{c_H y_H} = -\lg\dfrac{0.001 Ms}{\rho_{\text{sln}}}$ $= 1.74 + \lg\rho_{\text{sln}}$ | $pH_x - pH_c = -\lg\dfrac{f_H x_H}{c_H y_H} = -\lg\dfrac{0.001 Ms}{\rho_{solvent}}$ $= 1.74 + \lg\rho_{solvent}$ |

[a] Since Dr. Zuend suggested to use more geneanl equations (considering solvent other than water), the corrected equations also do not direcely appear in the revised mansucript. Instead they have been shown in a more general form as: $f_H = y_H 1000 \frac{dm^3}{m^3} \frac{M_w}{\rho_0} \frac{c_H}{x_H}$ (6) and $pH_x - pH_c = \log 10 \frac{1000 dm^3/m^3 M_s c^0}{\rho_0}$ (12) in the revised manuscript.

In the original manuscript, we discussed the effect of the density of aerosol solution on the conversion between molality-based pH and molarity-based pH, which formed the major part of the *Results and Discussion* section. However, after correction, the conversion between molarity and molality-based pH actually does not depend on the density of the solution but does so on the density of the pure solvent. Therefore the orignal discussion is not valid anymore. Consequently, we have revised the mansucript significantly. The major revisons include (1) removing the original discussion regarding comparision between $pH_m$ and $pH_c$; (2) incorporating the calculation of pH on differnet scales using multiple thermodynamci models; 3) using a gas+aerosol system instead of an aerosol only system to estimate pH; 4) discussing the differnce between different scales for a generalized solvent system instead of only water and (5) inclusion of a sub-section to discuss the general issues regarding aerosl acidity comparision across studies. Below are our detailed responses to each of the comments.

**Response to general comments:**
We have addressed Prof. Simon Clegg's comments point by point. A major revision of the manuscript has been conducted according to all comments. Especially, we have now revised the method to estimate aerosol acidity using a gas + aerosol system instead of the aerosol only system utilized in original version. We have also added an entire section (Section 3.2) in the revised manuscript to summarize the general issues for pH comparison across studies. A detailed response to each of the Reviewer's comments is listed below:

**2    Specific comments**

1. Abstract, line 16: Stating that this issue is addressed "for the first time" is rather bold given that the theoretical framework for activity coefficient and pH scale con- versions has been known for decades (even though it may be true that it is a frequently ignored issue, therefore it is worthy of attention by the atmospheric chemistry community).

   **Response:**
   We agree that the description here is inappropriate. We have deleted the concerned phrase in the description.

**Revised text:**

This study attempts to address this issue by comparing PM$_{2.5}$ aerosol pH based on different scales (molarity, molality and mole fraction) on the basis of theoretical considerations followed with a set of field data from Guangzhou, China as an example.

2. Abstract, line 18: "Using hourly ionic species measurements in Guangzhou, China, it is observed that $pH_x$ (mole fraction based) is always 1.74 pH unit higher than $pH_m$ (molality based)". This is clearly misleading, since the pH unit difference is not truly based on observation. It is a circular argument also made at other places in the manuscript. The 1.74 pH unit difference is in fact coming from the application of such a theory-based difference and using the same thermodynamic model to compute the pH values in different scales based on $H^+$ activity coefficient prediction from the same samples.

Response:
We agree with the reviewer's point that the relationship between $pH_x$ and $pH_m$ is from theory. The field data shown in this study is only an example of the application of the theory. The text has been revised as below.

**Revised text:**

This study attempts to address this issue by comparing PM$_{2.5}$ aerosol pH based on different scales (molarity, molality and mole fraction) on the basis of theoretical considerations followed with a set of field data from Guangzhou, China as an example. The three most widely used thermodynamic models (E-AIM-IV, ISORROPIA-II, and AIOMFAC) are employed for the comparison. It has been shown theoretically that the difference between pH$_x$ (mole fraction based) and pH$_m$ (molality based) is always a constant (1.74, when the solvent is water) within a thermodynamic model regardless of aerosol property.

3. Page 2, line 9: The authors state; "The acidity of aerosols can be quantified by parameters such as strong acidity, free acidity, cation-to-anion ratio and ammonium- to-sulfate ratio. However, these parameters neglect the effect of liquid water content or the dissociation of ions and acids (Pathak et al., 2004; Hennigan et al.,2015)." First, a definition of what strong acidity and free acidity represents is not given. Second, it is incorrect that free acidity neglects water content, as the partial dissociation of species like bisulfate is very much dependent on water content and therefore affects free acidity. Pathak et al. and Hennigan et al. do not seem to make such a point.

Response:
We agree that our original description was vague. In the revised manuscript, we have (1) added the definition of both strong and free acidity; and (2) specified the drawbacks of these parameters.

**Revised text:**

The most accurate parameter to characterize aerosol acidity is considered to be pH. The other parameters often used as proxies of aerosol acidity do not offer information on how acidic the particles are when they are present as aqueous droplets (Pathak et al., 2004). For example, strong acidity (defined as nmol of total H$^+$ per m$^3$ of air measured in water extracts of particles using the USEPA Reference Method (USEPA, 1992)) and ion charge balance are unable to

distinguish between free and undissociated H⁺ (e.g., protons associated with bisulfate) (Pathak et al., 2004;Hennigan et al., 2015). Ammonium-to-sulfate ratio and cation-to-anion ratio are unable to provide any measure of the degree of aerosol acidity even qualitatively (Hennigan et al., 2015). And lastly, free acidity (defined as the actual concentration of free H⁺ per m³ of air, not including the H⁺ released from bisulfate ions in aqueous extracts) represents the quantity of H⁺ in a specific volume of air while neglecting the concentration of H⁺ in liquid water (Pathak et al., 2004).

4. Equation (1): Define the meaning of "lg".

   Response:
   We have revised lg to log10 to make it clearer throughout the manuscript.

5. Equation (2): This equation and its description is flawed. (1) What are the terms of "1000" in the numerator and denominator? The authors likely use these for conversion from units of kg to g. If so, the mathematically correct way of writing this would be to write $1000\frac{g}{kg}$ in the equation and it would be necessary to state that the molar masses are supposed to be used in units of $\frac{g}{mol}$ rather than the standard SI unit of $\frac{kg}{mol}$. Otherwise it is simply incorrect and a potential source of confusion. However, since the ACP recommendation is to use SI units whenever possible, there is not need for these unit conversion terms at all (they would be 1). (2) This expression is only correct for the special case where the only solvent for the ions is water. However, in the more general case, there may be other solvents, such as organic compounds mixed with water and the ions in a liquid phase. In that case, the distinct molar masses of the organics must be accounted for in the activity coefficient conversion expression. Hence, since this is a key part of the discussion about different thermodynamic composition scales and reference and standard states, it should be shown correctly for the general case. A rigorous derivation of such scale conversions is, e.g., shown in the PhD thesis by Zuend (2007) (page 45 – 47 there), which shows different versions of the scale conversion formula. One of which (useful here) is $\ln[\gamma_i] = \ln[f_i^*] + \ln[\frac{x_i}{m_i M_w}]$ and f* the mole fraction scale activity coefficient, both with reference state of infinite dilution in pure water; $x_i$ the ion mole fraction with respect to dissociated ions, $m_i$ the ion molality and $M_w$ the molar mass of water. Since solvents other than water are also included in both the mole fraction and molality expressions for ion "i", this is a general expression. From this scale conversion of ion activity coefficients, it is readily shown that the difference in pH values is generally given by $pH_x - pH_m = -\log10[m^0 M_w]$, where m° denotes unit molality (= 1 mol/kg) (similar to Eq. (8) in the manuscript, but note the difference in units, the given Eq. (8) is not dimensionless in the log).

   Response:
   We thank the Reviewer for pointing out this issue.
   1) The factor of 1000 in the original equation was for the conversion of kg to g. The unit of molecular mass was g/mol in the original manuscript. We have now revised the unit of molecular mass to the SI unit of kg mol⁻¹ as suggested. All other units in the manuscript are SI units except molarity (mol dm⁻³). So $1000\frac{dm^3}{m^3}$ will be still shown when molarity of ion is involved.

2) We agree that the original Eq.2 is only valid when water is the only solvent. We have now cited Eq. 2.59 in the PhD thesis by Zünd (2007) and Robinson and Stokes (2002) to show the conversion (1) between $\gamma_H$ and $f_H$ (Eq. 4); (2) between $\gamma_H$ and $y_H$ (Eq. 4); and (3) between $f_H$ and $y_H$ in a more general form as below.

**Revised equations:**

$$\gamma_H = f_H \frac{x_H}{m_H M_s} \qquad (4)$$

$$\gamma_H = 1000 \frac{dm^3}{m^3} \frac{c_H y_H}{m_H \rho_0} \qquad (5)$$

$$f_H = y_H 1000 \frac{dm^3}{m^3} \frac{M_w}{\rho_0} \frac{c_H}{x_H} \qquad (6)$$

6. As pointed out in the comment by Simon Clegg, thermodynamic models differ in the way single-ion activity coefficients are calculated (since only mean cation/anion pair activity coefficients are measureable) and of course they also differ in the expressions, such that even with the correct conversion of activity coefficients or pH values between scales, different models may predict different $pH_m$ (or $pH_x$) values for the same input composition. Furthermore, only some models account for the influence of organic species in the mixture (e.g. AIOMFAC can be used for that, while ISORROPIA is only for inorganic aqueous mixtures) and differences in predicted pH may partially stem from organic interactions with ions and treatment of phase separation, see Pye et al (2018).

**Response:**
We agree with the Reviewer's point. This comment has been handled in combination with #10. We have now added Section 3.2 to discuss the general issues when comparing aerosol acidity across studies (including the difference in models, forward vs. reverse, stable vs. metastable, effect of non-volatile cations and effect of organic compounds).

**Revised text:**
It has been shown above that proper scale conversion has to be conducted when aerosol pH is compared. However, one should bear in mind that even with the same measured data and scale, pH calculated with different thermodynamic models or with different parameters may still not be comparable. Below, we briefly describe some of the general issues that need to be considered when aerosol acidity is compared across studies along with a summary of parameters used in the published studies in Table S1.

(1) Thermodynamic models like ISORROPIA-II and E-AIM can run in forward mode and reverse mode which result in significant difference (Song et al., 2018, Hennigan et al., 2015). It is recommended to use thermodynamic models in forward mode (gas plus aerosol as input) instead of reverse mode (aerosol only as input) which is highly sensitive to measurement uncertainties (Hennigan et al., 2015).

(2) Thermodynamic model can also be run in stable (liquid only) or metastable modes (both solid and liquid) which has not been specified in many studies (Table S1). Song et al. (2018) have shown that model calculations coupled with stable or metastable state assumptions can provide reasonable estimates of aerosol water pH. However, as pointed by Song et al. (2018), the studies

using standard ISORROPIA-II (without code correction) running in stable mode have predicted unrealistic pH values of around 7 and should be reevaluated.

(3) The effect of non-volatile cations such as $Na^+$, $Ca^{2+}$, $Mg^{2+}$ and $K^+$ on aerosol pH may also not be ignored. Although the effect of non-volatile cations on pH may be only 0.2-0.5 pH units, their impact on predicted partitioning of a semi-volatile species can be significant due to the highly non-linear response of $NH_3$-$NH_4^+$ or $HNO_3$-$NO_3^-$ partitioning to pH (Guo et al., 2017). Since E-AIM cannot explicitly treat $Ca^{2+}$, $Mg^{2+}$ and $K^+$ (unlike ISORROPIA-II and AIOMFAC), pH estimated using E-AIM may ignore $Ca^{2+}$, $Mg^{2+}$ and $K^+$ (as shown in Table S1) or treat them as equivalent sodium (e.g. (Hennigan et al., 2015)). Even if all non-volatile cations are treated as $Na^+$, the predicted thermodynamic states can be biased due to the strong non-ideality of divalent ions as well as variations in water uptake characteristics between $Na^+$ salts and its counterparts (e.g., Fountoukis et al., 2009).

(4) Most studies so far have estimated pH of aerosols with only inorganic compounds. However, it has been reported that pH can be affected by organic compounds in several ways. Guo et al., (2015) have shown that the pH can be increased by 0.15 to 0.23 units when aerosol water associated with organic compounds is considered. Omission of the contribution of organic acids to $H^+$ has been estimated to increase the pH by 0.07±0.03 by Song et al., (2018) using E-AIM-IV. It has been shown recently that accounting for non-ideal mixing can modify the pH such that a fully interactive inorganic–organic system showed a pH roughly 0.7 units higher than that predicted using an inorganic only system by AIOMFAC (Pye et al., 2018).

7. Equation (4): Similar to above comment. A general expression should be shown, with proper use of units.

**Response:**
Thanks. Equation 4 has been revised accordingly (Eq. 6 in the revised manuscript).

**Revised equations:**

$$f_H = y_H 1000 \frac{dm^3}{m^3} \frac{M_w}{\rho_0} \frac{c_H}{x_H} \qquad (6)$$

8. Page 4, line 3: the last sentence there makes little sense. The pH values can be compared when the scale effect is accounted for; the point is that one should not expect the values to be equal.

**Response:**
We assume that the Reviewer actually meant page 5 in the original manuscript. This sentence has been deleted in the revised manuscript.

9. Page 4, line 13: "This is supported by our field data". Again, this is a circular argument. The $pH_x - pH_m$ scale difference is used in the evaluation of the pH values, so of course it will show as consistent, but the measurements are no proof for that. Also, the fact that the difference should be a constant in pH units is clear from the theory, as long as the same thermodynamic model is used to compute the activity coefficients (which may not be the case when different studies are compared).

**Response:**

We assume the Reviewer actually meant page 5 in the original manuscript. We agree with the reviewer's point and have revised the text accordingly. As we clarified in the beginning of our response, other revisions have also been made and the final revised text is as below.

**Revised text:**

The difference of $pH_x$ and $pH_m$ is $\log 10 M_w m^0$ (according to Eq. 10) which is only determined by the molecular weight of the solvent. When water is the only solvent in the system (molecular weight of 0.018 kg mol$^{-1}$), $pH_x$ - $pH_m$ is fixed at 1.74 within the model regardless of aerosol property or the model (as in this study).

10. Page 7, line 6: The discussion in this paragraph is not sufficient to address the other very important issues when aerosol acidity is attempted to be assessed from field measurements. It is also clear from theory and comparisons that organic compounds will affect aerosol acidity, maybe not dramatically but noticeably, since their interactions with water and $H^+$ ions are affecting the activity of $H^+$. Last sentence in paragraph: "The relationship between pHx, pHc and pHm established in this study is valid regardless of the method selected to estimate aerosol acidity." This is true only for the scale conversion, since it depends on proper application of thermodynamic theory only (which has been known for decades and is not a novelty of this study). However, when aerosol sample acidity is estimated in practice, different models are used for activity coefficients (e.g. ISORROPIA, E-AIM, AIOMFAC) or unit activity coefficients are assumed (not recommended). Moreover, different methods are applied to determine the approximate H+ amount, which is a critical problem in acidity evaluation, see Hennigan et al (2015). Therefore, the difference in reported pH values is not just due to the offsets between these different pH scales. The authors have failed to make this important point very clear.

**Response:**

We agree with the reviewer's point. This comment has been handled in combination with #6. We have now added Section 3.2 to discuss the general issues when comparing aerosol acidity across studies (including difference in models, forward vs. reverse, stable vs. metastable, effect of non-volatile cations and effect of organic compounds). The revised text is shown in the response to comment #6.

11. Table 1: The definitions include many mistakes and typos; e.g., pH$_m$ is not molarity based, the last two entries are confusing and not correctly described and reference states of activity coefficients are missing.

**Response:**

Table 1 has been revised thoroughly. The reference state of activity and activity coefficient have been added in the definition of activity. The revised Table 1 is shown below.

**Revised table:**

**Table 1. List of abbreviations.**

| Abbreviation | Definition |
|---|---|
| $ac_H$ | activity of hydrogen ions with standard state of the hypothetical ideal aqueous solution of unit molarity and reference state of infinite dilute solution (dimensionless) |
| $am_H$ | activity of hydrogen ions with standard state of the hypothetical ideal aqueous solution of unit molality and reference state of infinite dilute solution (dimensionless) |
| $ax_H$ | activity of hydrogen ions with standard state of the hypothetical pure $H^+$ solution and reference state of infinite dilute solution (dimensionless) |
| $c^0$ | unit molarity (1 mol dm$^{-3}$ solution) |
| $c_H$ | molarity of hydrogen ion (mol dm$^{-3}$ solution) |
| $c_i$ | molarity of solute species $i$ (mol dm$^{-3}$ solution) |
| $f_H$ | mole fraction scale activity coefficient |
| $m^0$ | unit molality (1 mol kg$^{-1}$ solvent) |
| $m_H$ | molality of hydrogen ions (mol kg$^{-1}$ solvent) |
| $m_i$ | molality of solute species $i$ (mol kg$^{-1}$) |
| $M_i$ | molecular mass of solute species $i$ (kg mol$^{-1}$) |
| $M_s$ | molecular weight of single solvent or averaged molecular weight for multiple solvents (kg mol$^{-1}$ ) |
| $pH_c$ | molarity based pH (dimensionless) |
| $pH_m$ | molality based pH (dimensionless) |
| $pH_x$ | mole fraction based pH (dimensionless) |
| $x_H$ | mole fraction of hydrogen ions (dimensionless, mol $H^+$ in total moles) |
| $y_H$ | molarity scale activity coefficient |
| $\gamma_H$ | molality scale activity coefficient |
| $\rho_0$ | density of pure solvent (kg m$^{-3}$) |
| $\rho_{sln}$ | density of hygroscopic aerosol solution (kg m$^{-3}$) |

**Reference**

USEPA, 1992. Determination of strong acidity of atmospheric fine-particles (<2.5 μm) using annular denuder technology. Atmospheric Research and Exposure Assessment Laboratory, EPA Report No. EPA/600/R-93/037, Washington, DC.

Fountoukis, C., Nenes, A., Sullivan, A., Weber, R., Van Reken, T., Fischer, M., Matías, E., Moya, M., Farmer, D., and Cohen, R. C.: Thermodynamic characterization of Mexico City aerosol during MILAGRO 2006, Atmos. Chem. Phys., 9, 2141-2156, 10.5194/acp-9-2141-2009, 2009.

Guo, H., Nenes, A., and Weber, R. J.: The underappreciated role of nonvolatile cations on aerosol ammonium-sulfate molar ratios, Atmos. Chem. Phys. Discuss., 2017, 1-19, 10.5194/acp-2017-737, 2017.

Hennigan, C. J., Izumi, J., Sullivan, A. P., Weber, R. J., and Nenes, A.: A critical evaluation of proxy methods used to estimate the acidity of atmospheric particles, Atmos. Chem. Phys., 15, 2775-2790, 10.5194/acp-15-2775-2015, 2015.

Pathak, R. K., Louie, P. K. K., and Chan, C. K.: Characteristics of aerosol acidity in Hong Kong, Atmos. Environ., 38, 2965-2974, https://doi.org/10.1016/j.atmosenv.2004.02.044, 2004.

Pye, H. O. T., Zuend, A., Fry, J. L., Isaacman-VanWertz, G., Capps, S. L., Appel, K. W., Foroutan, H., Xu, L., Ng, N. L., and Goldstein, A. H.: Coupling of organic and inorganic aerosol systems and the effect on gas–particle partitioning in the southeastern US, Atmos. Chem. Phys., 18, 357-370,

10.5194/acp-18-357-2018, 2018.

Robinson, R. A., and Stokes, R. H.: Electrolyte solutions, Courier Corporation, 2002.

Song, S., Gao, M., Xu, W., Shao, J., Shi, G., Wang, S., Wang, Y., Sun, Y., and McElroy, M. B.: Fine-particle pH for Beijing winter haze as inferred from different thermodynamic equilibrium models, Atmos. Chem. Phys., 18, 7423-7438, 10.5194/acp-18-7423-2018, 2018.

van Boekel, M. A. J. S.: Kinetic Modeling of Reactions In Foods, CRC Press, New York, 2008.

Zünd, A.: Modelling the thermodynamics of mixed organic-inorganic aerosols to predict water activities and phase separations, ETH Zurich, 2007.

---

## Author Comment (AC2) · 26 Jun 2018

**Response to Anonymous Referee #2**

*Comments are in black, responses in* *blue* *and the revised text in* *red.*

Particle acidity is an important property in terms of aerosol chemistry and its impact on health and climate. Due to the importance, there has been an increasing amount of publication investigating particle pH through thermodynamic models in recent years, which is a step forward compared to the problematic ion balance or molar ratio because of particle liquid water and non-ideality effect (activity coefficient). Without a consensus on the pH definition, it would be difficult to compare various studies, and this may lead to misunderstanding or misinterpretation of particle pH. Although some particle acidity studies state clearly the pH definition used in the analyses, there are some papers not talking about the definition at all. In the latter case, some studies probably take the approach of molality-based or molarity-based because they take a pH of 7 as neutral.

This paper shows the two pHs are very close and the minor difference is caused by particle density (for dilute water solution, no difference is expected from the two pHs). In general, this paper hits an important point (i.e., pH definition) that has not been paid enough attention to by the community and fits the scope of a technical note on ACP. However, some revisions are needed before considering a publication.

**Response:**
We thank the Reviewer for the helpful comments and suggestions, and also for agreeing with the importance of pH interconversion between different scales, which is the main focus of current study.

First of all, we would like to apologize for the mistakes made in Eqs. 4 and 9 in the original manuscript that stem from errors originally made in the equation by van Boekel (2008) used in our reference. Since the mistakes are not included in either of the reviewer's comments or in the short comments by Prof. Clegg, we clarify this issue first before responding to the reviewers' comments in detail. The details of the mistakes are shown in Table 1 below.

Table 1. Correction of Eqs. 4 and 9 in the original manuscript

| Eq. # | Original (with mistake) | Corrected [a] |
|---|---|---|
| 4 | $y_H = f_H \dfrac{\rho_{sln}}{\rho_{sln} + 0.001[M_s\sum c_i - \sum c_i M_i]}$ | $y_H = f_H \dfrac{\rho_{solvent}}{\rho_{sln} + 0.001[M_s\sum c_i - \sum c_i M_i]}$ |
| 9 | $pH_x - pH_c = -\lg\dfrac{f_H x_H}{c_H y_H} = -\lg\dfrac{0.001 Ms}{\rho_{sln}}$ $= 1.74 + \lg\rho_{sln}$ | $pH_x - pH_c = -\lg\dfrac{f_H x_H}{c_H y_H} = -\lg\dfrac{0.001 Ms}{\rho_{solvent}}$ $= 1.74 + \lg\rho_{solvent}$ |

[a] Since Dr. Zuend (Referee 1) suggested to use more genearl equations (considering solvent other than water), the corrected equations also do not direcely appear in the revised mansucript. Instead they have been shown in a more general form as: $f_H = y_H 1000 \dfrac{dm^3}{m^3} \dfrac{M_w}{\rho_0} \dfrac{c_H}{x_H}$ (6) and $pH_x - pH_c = \log 10 \dfrac{1000 dm^3/m^3 M_s c^0}{\rho_0}$ (12) in the revised manuscript.

In the original manuscript, we discussed the effect of the density of aerosol solution on the conversion between molality-based pH and molarity-based pH, which formed the major part of the *Results and Discussion* section. However, after correction, the conversion between molarity and molality-based pH actually does not depend on the density of the solution but does so on the density of the pure solvent. Therefore the orignal discussion is not valid anymore. Consequently, we have revised the mansucript significantly. The major revisons include (1) removing the original discussion regarding comparision between $pH_m$ and $pH_c$; (2) incorporating the calculation of pH on differnet scales using multiple thermodynamci models; 3) using a gas+aerosol system instead of an aerosol only system to estimate pH; 4) discussing the differnce between different scales for a generalized solvent system instead of only water and (5) inclusion of a sub-section to discuss the general issues regarding aerosl acidity comparison across studies. Below are our detailed responses to each of the comments.

1. There are several "circular" statements as pointed out by the other reviewer Andreas Zuend. The differences between pHx, pHc, pHm are all expected from their definitions. This is the key point of this paper: the difference between pHc and pHm is small (within 0.2 pH units, caused by particle density), but pHx is significantly higher than pHc and pHm (pHx – pHm = 1.74). The E-AIM (or any other model) predicted pHs are supposed to be consistent with the rule, if one model is used consistently in this analysis. Therefore, it is not a real support by field data (used as E-AIM input) as claimed on line 13 Page 5. Relating to this, the author should consider changing the saying of "observed" (Line 18 Page 1) in the abstract. The presented result is all based on a thermodynamic model prediction of pH but not measured pH.

   **Response:**
   We agree with the Reviewer's point. The differences between $pHx$, $pHc$ and $pHm$ are expected from their definitions. The field data shown in this study is actually an example to show the conversion between different scales. The following revision has been done.

   **Revised text**
   This study attempts to address this issue by comparing $PM_{2.5}$ aerosol pH based on different scales (molarity, molality and mole fraction) on the basis of theoretical considerations followed with a set of field data from Guangzhou, China as an example.

2. From a boarder view of an application, this paper could be more beneficial by showing the default pH scales given by widely used thermodynamic models. The E-AIM model has been discussed in detail, however, ISORROPIA or AIOMFAC or any other model is not mentioned. For example, ISORROPIA gives pH in the model output and the pH scale is molality-based (Fountoukis et al., 2009; Guo et al., 2015). If a literature doesn't specify the pH scale, this piece of info could be very helpful to readers.

   **Response:**
   We agree with the Reviewer's point. Following the reviewer's suggestion, we have now shown the calculations with all the 3 models. We have also described the parameters that can be

obtained from the model outputs as well as how other parameters are estimated. The revised text is shown below.

**Revised text:**

Table S1. A summary of estimation methods of parameters for pH calculation based on different standard states.

| Parameters | E-AIM-IV | ISORROPIA-II | AIOMFAC |
|---|---|---|---|
| **Mole fraction** | | | |
| $x_H$ | output | Eq. (7) | output |
| $f_H$ | output | 1[a] | Eq. (4) |
| **Molality** | | | |
| $m_H$ | output | output | output |
| $\gamma_H$ | Eq. (4) | 1[a] | output |
| **Molarity** | | | |
| $c_H$ | Eq. (8) [b] | Eq. (8) [b] | Eq. (8) [b] |
| $y_H$ | Eq. (6) | 1[a] | Eq. (6) |

Note: [a] activity coefficient is assumed to be 1; [b] the density of aerosol solution is based on the result from E-AIM-IV.

3. The section 3.3 is problematic without discussion of uncertainty, especially considering that the presented pH in this study is solely based on particle data (no gas data used to constrain pH). The predicted pH uncertainty is propagated from the particle ionic composition data (6% reported by Chen et al. (2016)) and RH, T. It is not easy to estimate particle pH uncertainty. Guo et al. (2015) estimated a pH uncertainty of 13% using another model, ISORROPIA, and in forward model for their dataset. Even though the two data points (S-I and S-II) are selected for the largest deviation from the1:1 line, small differences (7% or 8%) in hydrogen ion activity are found, which seem to be within uncertainty range. Hennigan et al. (2015) and other papers have pointed out that forward mode is superior over reverse mode in terms of particle pH prediction accuracy. For one reason, reverse mode is more sensitive to particle measurement uncertainty (likely the cause for occasionally very high pH seen in Figure 1). For the other reason, particle pH is sensitive to gas-particle partitioning of semivolatile species (e.g., NH4+, NO3-, and Cl-), as long as the species is not totally in gas phase or particle phase. The forward mode predicted pH can then be validated if predicted and observed gas-particle partitioning agree. In contrast, the reverse mode predicted pH could not be verified in the same way. The Line 7 on Page 7 seems to suggest there is no advantage of using forward mode calculation and this is misleading to potential users.

**Response:**
We thank the reviewer for pointing out this important issue.
(1) We have revised the method to calculate aerosol pH from the forward mode, which is of lower uncertainty, instead of reverse mode.
(2) We acknowledge that the difference between $pH_m$ and $pH_c$ may not be large compared with uncertainties from measurement errors. Therefore, we have added the following text in the discussion.

**Revised text:**
Given that the uncertainty of pH calculation due to measurement errors can be as high as 14% (Guo et al., 2015), the difference of $pH_x$ and $pH_m$ can simply fall within the range of measurement errors. However, the bias between $pH_x$ and $pH_m$ can be considered to be a systematic one, which needs to be addressed for the sake of comprehensiveness in theoretical analysis. Moreover, even small biases in pH may imply substantial partitioning errors for semivolatile species like ammonium, nitrate, chloride, and even organic acids (Guo et al., 2017).

4. As Andreas Zuend points out, it would be nice to mention the possible differences be- tween predicted pH via different models at the end of section 3.4. Even if the same inputs are given to models, 100% agreement in pH is not expected due to differences in assumptions and approaches. There are a number of studies comparing thermodynamic models and exploring the differences (Hennigan et al., 2015; Liu et al., 2017; Pye et al., 2018; Song et al., 2018)

**Response:**

We agree with the Reviewer's point. We have addressed this issue in the revised manuscript.

(1) Following the reviewer's suggestion, we have shown the comparison of pH calculated using different thermodynamic models in the revised manuscript as below followed by a comparison of the three models.

[revised manuscript text omitted]

highly non-linear response of $NH_3$-$NH_4^+$ or $HNO_3$-$NO_3^-$ partitioning to pH (Guo et al., 2017). Since E-AIM cannot explicitly treat $Ca^{2+}$, $Mg^{2+}$ and $K^+$ (unlike ISORROPIA-II and AIOMFAC), pH estimated using E-AIM may ignore $Ca^{2+}$, $Mg^{2+}$ and $K^+$ (as shown in Table S1) or treat them as equivalent sodium (e.g. (Hennigan et al., 2015)). Even if all non-volatile cations are treated as $Na^+$, the predicted thermodynamic states can be biased due to the strong non-ideality of divalent ions as well as variations in water uptake characteristics between $Na^+$ salts and its counterparts (e.g., Fountoukis et al., 2009).

(4) Most studies so far have estimated pH of aerosols with only inorganic compounds. However, it has been reported that pH can be affected by organic compounds in several ways. Guo et al., (2015) have shown that the pH can be increased by 0.15 to 0.23 units when aerosol water associated with organic compounds is considered. Omission of the contribution of organic acids to $H^+$ has been estimated to increase the pH by 0.07±0.03 by Song et al., (2018) using E-AIM-IV. It has been shown recently that accounting for non-ideal mixing can modify the pH such that a fully interactive inorganic–organic system showed a pH roughly 0.7 units higher than that predicted using an inorganic only system by AIOMFAC (Pye et al., 2018).

**Minor comments:**

1.  Toning down the statement of "for the first time" is suggested. Thermodynamic specialists must have known the difference between pH scales. The elucidation in the paper is based on established equations on textbooks and doesn't sound to me like a groundbreaking finding. The paper is of value to minimize the gaps in the current understanding of pH definitions.

**Response:**
Thanks for pointing out the issue. We have deleted the concerned phrase in the description and elsewhere in the revised manuscript.

**Revised text:**
This study attempts to address this issue by comparing $PM_{2.5}$ aerosol pH based on different scales (molarity, molality and mole fraction) on the basis of theoretical considerations followed with a set of field data from Guangzhou, China as an example.

2.  The units in Equation (3) (mol/kg water) and (6) (mol/dm3) look redundant. However, more explanations in the text are needed. Equation (3) is defined based on the mass of water, while Equation (6) is defined based on the volume of particle, which includes the volumes of water (solvent) and other solutes. If Equation (6) is defined solely on the volume of particle water, there would not be any difference with Equation (3). This is not clear in the text.

**Response:**
We thank the reviewer pointing out the issue. The two equations have been changed accordingly. We have also explained that molarity means mol $dm^{-3}$ solution while molality means mol $kg^{-1}$ solvent in Table 1.

**Revise equations:**
$$pH_c = -\log 10(a_{c_H}) = -\log 10\left(\frac{y_H c_H}{c^o}\right) \qquad (2)$$

$$pH_m = -\log 10(a_{m_H}) = -\log 10(\frac{\gamma_H m_H}{m^o}) \qquad (3)$$

**Explanation of parameters (in Table 1):**

| | |
|---|---|
| $m_H$ | molality of hydrogen ions (mol kg$^{-1}$ solvent) |
| $c_H$ | molarity of hydrogen ion (mol dm$^{-3}$ solution) |

3.  Proper reference as suggested by Simon Clegg.

**Response:**
We have now added the sources of all equations as suggested by Prof. Simon Clegg in Table 2.

**Revised text:**
Table 2. Summary of equations for the interconversion of concertation and activity coefficient based on different standard states.

| Parameter | $pH_x$ vs. $pH_m$ | | $pH_m$ vs. $pH_c$ | | $pH_x$ vs. $pH_c$ | |
|---|---|---|---|---|---|---|
| Activity coefficient [a] | $\gamma_H = f_H \dfrac{x_H}{m_H M_s}$ | (4) | $\gamma_H = 1000\dfrac{dm^3}{m^3}\dfrac{c_H y_H}{m_H \rho_0}$ | (5) | $f_H = y_H 1000\dfrac{dm^3}{m^3}\dfrac{M_w}{\rho_0}\dfrac{c_H}{x_H}$ (6) | |
| Concentration [b] | $x_H = \dfrac{m_H}{\sum v_i m_i + \frac{1}{M_s}}$ | (7) | $c_H = \dfrac{m_H}{\frac{\sum v_i m_i M_i + 1}{\rho_{sln}}}$ | (8) | $x_H = \dfrac{M_s c_H}{M_s \sum c_i + 0.001\frac{m^3}{dm^3}\rho_{sln} - \sum c_i M_i}$ (9) | |
| pH [c] | $pH_x - pH_m = -\log 10 M_s m^0$ (10) | | $pH_m - pH_c = -\log 10\frac{c^0 1000 dm^3/m^3}{m^0 \rho_0}$ (11) | | $pH_x - pH_c = \log 10\frac{1000 dm^3/m^3 M_s c^0}{\rho_0}$ (12) | |

Note: [a] Source of Eqs. (4)-(5) are Robinson and Stokes (2002) and the source of Eq. (6) is Zünd (2007). The details of derivation of Eqs. (4)-(6) are shown in Robinson and Stokes (2002) and Zünd (2007). [b] Eqs. (7) - (9) are based on the definition of each parameter. [c] Eqs. (10)-(12) are derived from Eqs (4)-(6) and (7)-(9) for each standard state.

---

## Author Comment (AC3) · 26 Jun 2018

**Response to Prof. Clegg**

*Comments are in black, responses in* blue *and the revised text in* red.

I believe I reviewed a very similar ms to this one (submitted to another journal) in 2017. I presume it was rejected.

Below are what I see as some of the most important issues regarding our understanding (from measurement or from models) of aerosol acidity. I don't think that the authors really grasp them or the problems involved, and that this proposed Technical Note does not provide the clarity and guidance that are needed by the field.

**Response:**

We thank Prof. Simon Clegg for the helpful comments.

First of all, we would like to apologize for the mistakes made in Eqs. 4 and 9 in the original manuscript that stem from errors originally made in the equation by van Boekel (2008) used in our reference. Since the mistakes are not included in either of the reviewer's comments or in the short comments by Prof. Clegg, we clarify this issue first before responding to the reviewers' comments in detail. The details of the mistakes are shown in Table 1 below.

Table 1. Correction of Eqs. 4 and 9 in the original manuscript

| Eq. # | Original (with mistake) | Corrected [a] |
|---|---|---|
| 4 | $y_H = f_H \dfrac{\rho_{\text{sln}}}{\rho_{\text{sln}} + 0.001[M_s\sum c_i - \sum c_i M_i]}$ | $y_H = f_H \dfrac{\rho_{solvent}}{\rho_{\text{sln}} + 0.001[M_s\sum c_i - \sum c_i M_i]}$ |
| 9 | $pH_x - pH_c = -\lg\dfrac{f_H x_H}{c_H y_H} = -\lg\dfrac{0.001 Ms}{\rho_{\text{sln}}}$ $= 1.74 + \lg\rho_{\text{sln}}$ | $pH_x - pH_c = -\lg\dfrac{f_H x_H}{c_H y_H} = -\lg\dfrac{0.001 Ms}{\rho_{solvent}}$ $= 1.74 + \lg\rho_{solvent}$ |

[a] Since Dr. Zuend (Referee 1) suggested to use more genearl equations (considering solvent other than water), the corrected equations also do not direcely appear in the revised mansucript. Instead they have been shown in a more general form as: $f_H = y_H 1000\dfrac{dm^3}{m^3}\dfrac{M_w}{\rho_0}\dfrac{c_H}{x_H}$  (6) and $pH_x - pH_c = \log 10 \dfrac{1000 dm^3/m^3 M_s c^0}{\rho_0}$ (12) in the revised manuscript.

In the original manuscript, we discussed the effect of the density of aerosol solution on the conversion between molality-based pH and molarity-based pH, which formed the major part of the *Results and Discussion* section. However, after correction, the conversion between molarity and molality-based pH actually does not depend on the density of the solution but does so on the density of the pure solvent. Therefore the orignal discussion is not valid anymore. Consequently, we have revised the mansucript significantly. The major revisons include (1) removing the original discussion regarding comparision between $pH_m$ and $pH_c$; (2) incorporating the calculation of pH on differnet scales using multiple thermodynamci models;  3) using a gas+aerosol system instead of an aerosol only system to estimate pH; 4) discussing the differnce between different scales in a more general solvent instead of water only and (5) inclusion of a sub-section to discuss the general issues regarding aerosl acidity comparision across studies. Below are our detailed responses to each of the comments.

**Response to general comment:**

We agree that the clarity needs to be improved. We have revised the manuscript based on these and the other two reviewers' comments. We need to point out that we do not intend to provide a fundamental and comprehensive understanding of pH in this study. Instead, we have focused on the interconversion of pH between different scales as this is an important but often overlooked issue. Therefore, other issues regarding aerosol acidity which are well understood or well-practiced in the community are not discussed in the current study.

1. Any discussion of pH ought to start with a recognition of what it is, which is not some "absolute" but depends on either how it is measured or the model used to calculate it. See for example the first sentence of Buck et al. (Pure Appl. Chem. 74, 2169-2200, 2002): "The concept of pH is unique among the commonly encountered physicochemical quantities listed in the IUPAC Green Book in that, in terms of its definition, pH = -lg a it involves a single ion quantity, the activity of the hydrogen ion, which is immeasurable by any thermodynamically valid method and requires a convention for its evaluation." So, for example, values of model-calculated H+ activities depend on how the model "splits" mean activity coefficients of cation-anion pairs (which are measurable) into single ion activities (which are not). This many not be a problem if all the calculations of thermodynamic properties (gas/aerosol equilibrium for example) are made with one model, but it should nonetheless be recognised. The measurement of pH is tied to the defined pH of the buffer used to calibrate the instrument. The buffer needs to be chemically similar to the solution being measured. Measured and modelled pH may not be comparable.

**Response:**

We agree with this point. We did not discuss the fundamentals of pH measurement or estimation in our original manuscript because we simply focused on the scale conversion.

    (1) To make the manuscript more comprehensive and clearer, we have now added a relevant description in the introduction.

**Revised text:**
As per the International Union of Pure and Applied Chemistry (IUPAC), pH is defined as the negative log activity of hydrogen ions (https://goldbook.iupac.org/html/P/P04524.html). It is immeasurable because its definition involves a single ion quantity, the hydrogen ion activity (Baucke, 2002). Therefore, the value of pH is not an absolute one but depends on either how it is measured or the model used to calculate it. Especially, for aerosol pH, a commonly accepted measurement method is lacking despite some recent developments (Rindelaub et al., 2016), and it is usually calculated from thermodynamic models in practice.

    (2) We have also added a section to discuss the general issues related to aerosol acidity comparison besides scale conversion.

**Revised text:**
  3.2 General issues with pH comparison
  It has been shown above that proper scale conversion has to be conducted when aerosol pH is compared. However, one should bear in mind that even with the same measured data and scale, pH calculated with different thermodynamic models or with different parameters may still not be comparable. Below, we briefly describe some of the general issues that need to be considered when aerosol acidity is compared across studies along with a summary of parameters used in the published studies in Table S1.

(1) Thermodynamic models like ISORROPIA-II and E-AIM can run in forward mode and reverse mode which result in significant difference (Song et al., 2018, Hennigan et al., 2015). It is recommended to use thermodynamic models in forward mode (gas plus aerosol as input) instead of reverse mode (aerosol only as input) which is highly sensitive to measurement uncertainties (Hennigan et al., 2015).

(2) Thermodynamic model can also be run in stable (liquid only) or metastable modes (both solid and liquid) which has not been specified in many studies (Table S1). Song et al. (2018) have shown that model calculations coupled with stable or metastable state assumptions can provide reasonable estimates of aerosol water pH. However, as pointed by Song et al. (2018), the studies using standard ISORROPIA-II (without code correction) running in stable mode have predicted unrealistic pH values of around 7 and should be reevaluated.

(3) The effect of non-volatile cations such as $Na^+$, $Ca^{2+}$, $Mg^{2+}$ and $K^+$ on aerosol pH may also not be ignored. Although the effect of non-volatile cations on pH may be only 0.2-0.5 pH units, their impact on predicted partitioning of a semi-volatile species can be significant due to the highly non-linear response of $NH_3$-$NH_4^+$ or $HNO_3$-$NO_3^-$ partitioning to pH (Guo et al., 2017). Since E-AIM cannot explicitly treat $Ca^{2+}$, $Mg^{2+}$ and $K^+$ (unlike ISORROPIA-II and AIOMFAC), pH estimated using E-AIM may ignore $Ca^{2+}$, $Mg^{2+}$ and $K^+$ (as shown in Table S1) or treat them as equivalent sodium (e.g. (Hennigan et al., 2015)). Even if all non-volatile cations are treated as $Na^+$, the predicted thermodynamic states can be biased due to the strong non-ideality of divalent ions as well as variations in water uptake characteristics between $Na^+$ salts and its counterparts (e.g., Fountoukis et al., 2009).

(4) Most studies so far have estimated pH of aerosols with only inorganic compounds. However, it has been reported that pH can be affected by organic compounds in several ways. Guo et al., (2015) have shown that the pH can be increased by 0.15 to 0.23 units when aerosol water associated with organic compounds is considered. Omission of the contribution of organic acids to $H^+$ has been estimated to increase the pH by $0.07\pm0.03$ by Song et al., (2018) using E-AIM-IV. It has been shown recently that accounting for non-ideal mixing can modify the pH such that a fully interactive inorganic–organic system showed a pH roughly 0.7 units higher than that predicted using an inorganic only system by AIOMFAC (Pye et al., 2018).

2. The H+ content of an aerosol should never be calculated from charge balance, for reasons that are so obvious that they hardly need repeating here (chemical analysis may not include all charged species; real H+ content is likely to be small relative to the amounts of the other cations and anions and is obtained as the [small] difference between two large and uncertain numbers). The use of measurements of gas phase NH3 or HNO3, in addition to the major ion composition of the aerosol, as constraints on aerosol acidity seems the sensible approach to me. I'm aware that the group of Jose Jimenez uses this method. I expect there are others.

**Response:**

We agree that thermodynamic models constrained by gas + aerosol measurements are more accurate than aerosol only systems that are highly sensitive to measurement uncertainties as reported by Hennigan et al. (2015). We have revised the calculation of aerosol pH in current study to gas + aerosol measurements (forward mode) in the revised manuscript.

3. The relationship between solute activities on different concentration scales - the basis of equations 1 to 10 of the authors - is textbook material. See for example Chapter 2 of "Electrolyte Solutions" by Robinson and Stokes. The authors haven't cited this or any other chemistry textbook.

**Response:**

Thanks. All equations have now been compiled in Table 3 in the revised manuscript along with sources in the notes. For the interconversion of activity between different scales (Eq. 3 and 4), the book by Robinson and Stokes (2002) has been cited.

**Revised text:**
Table 3. Summary of equations for the interconversion of concertation and activity coefficient based on different standard states.

| Parameter | $pH_x$ vs. $pH_m$ | | $pH_m$ vs. $pH_c$ | | $pH_x$ vs. $pH_c$ | |
|---|---|---|---|---|---|---|
| Activity coefficient [a] | $\gamma_H = f_H \dfrac{x_H}{m_H M_S}$ | (4) | $\gamma_H = 1000 \dfrac{dm^3}{m^3} \dfrac{c_H y_H}{m_H \rho_0}$ | (5) | $f_H = y_H 1000 \dfrac{dm^3}{m^3} \dfrac{M_w}{\rho_0} \dfrac{c_H}{x_H}$ | (6) |
| Concentration [b] | $x_H = \dfrac{m_H}{\sum v_i m_i + \frac{1}{M_S}}$ | (7) | $c_H = \dfrac{m_H}{\frac{\sum v_i m_i M_i + 1}{\rho_{sln}}}$ | (8) | $x_H = \dfrac{M_S c_H}{M_S \sum c_i + 0.001 \frac{m^3}{dm^3} \rho_{sln} - \sum c_i M_i}$ | (9) |
| pH [c] | $pH_x - pH_m = -\log 10 [M_s m^0]$ | (10) | $pH_m - pH_c = -\log 10 \dfrac{c^0 1000 dm^3/m^3}{m^0 \rho_0}$ | (11) | $pH_x - pH_c = \log 10 \dfrac{1000 dm^3/m^3 M_s c^0}{\rho_0}$ | (12) |

Note: [a] Source of Eqs. (4)-(5) are Robinson and Stokes (2002) and the source of Eq. (6) is Zünd (2007). The details of derivation of Eqs. (4)-(6) are shown in Robinson and Stokes (2002) and Zünd (2007). [b] Eqs. (7)-(9) are based on the definition of each parameter. [c] Eqs. (10)-(12) are derived from Eqs (4)-(6) and (7)-(9) for each standard state.

4. The rough correlation between pH on the molarity scale and on the other scales is to be expected, as is clear from eq (9), and this part of their discussion doesn't really add to the content of the ms.

Response:
We agree that the correlation between pH on the molarity scale and on the other scales is to be expected solely based on theory. However, the discussion of field data here is still helpful since it shows that the conversion is related to the property of each sample. We can demonstrate how the conversion will be affected by the property of samples. For example, we observe a different trend of $pH_m$ and $pH_c$ due to different properties of the aerosol (when activity coefficient is assumed as unity in the revised manuscript). Therefore, we have kept the discussion in our revised manuscript (but corrected the mistakes). But as also pointed out by the other reviewers, our original statement in the abstract is not proper regarding the role of field data in this study. It has been revised accordingly in the abstract.

**Revised text:**
This study attempts to address this issue by comparing PM$_{2.5}$ aerosol pH based on different scales (molarity, molality and mole fraction) on the basis of theoretical considerations followed with a set of field data from Guangzhou, China as an example.

5. Discussions of aerosol acidity need to address the fact that "pH" will go up and down with the diurnal cycle of RH, while the H+ content of the aerosol might remain more or less the same.

Response:

We agree that if H$^+$ remains the same, pH will fluctuate with the diurnal cycle mainly due to the change of RH. However, this point is beyond the scope of the current study, and has been discussed extensively in other studies. (e.g. (Guo et al., 2015;Jia et al., 2018)) Therefore this is not discussed in current study.

**Reference**

Baucke, F. G.: New IUPAC recommendations on the measurement of pH – background and essentials, Anal. Bioanal. Chem., 374, 772-777, 10.1007/s00216-002-1523-4, 2002.

Fountoukis, C., Nenes, A., Sullivan, A., Weber, R., Van Reken, T., Fischer, M., Matías, E., Moya, M., Farmer, D., and Cohen, R. C.: Thermodynamic characterization of Mexico City aerosol during MILAGRO 2006, Atmos. Chem. Phys., 9, 2141-2156, 10.5194/acp-9-2141-2009, 2009.

Guo, H., Xu, L., Bougiatioti, A., Cerully, K. M., Capps, S. L., Hite Jr, J. R., Carlton, A. G., Lee, S. H., Bergin, M. H., Ng, N. L., Nenes, A., and Weber, R. J.: Fine-particle water and pH in the southeastern United States, Atmos. Chem. Phys., 15, 5211-5228, 10.5194/acp-15-5211-2015, 2015.

Guo, H., Nenes, A., and Weber, R. J.: The underappreciated role of nonvolatile cations on aerosol ammonium-sulfate molar ratios, Atmos. Chem. Phys. Discuss., 2017, 1-19, 10.5194/acp-2017-737, 2017.

Hennigan, C. J., Izumi, J., Sullivan, A. P., Weber, R. J., and Nenes, A.: A critical evaluation of proxy methods used to estimate the acidity of atmospheric particles, Atmos. Chem. Phys., 15, 2775-2790, 10.5194/acp-15-2775-2015, 2015.

Jia, S., Sarkar, S., Zhang, Q., Wang, X., Wu, L., Chen, W., Huang, M., Zhou, S., Zhang, J., Yuan, L., and Yang, L.: Characterization of diurnal variations of PM2.5 acidity using an open thermodynamic system: A case study of Guangzhou, China, Chemosphere, 202, 677-685, https://doi.org/10.1016/j.chemosphere.2018.03.127, 2018.

Pye, H. O. T., Zuend, A., Fry, J. L., Isaacman-VanWertz, G., Capps, S. L., Appel, K. W., Foroutan, H., Xu, L., Ng, N. L., and Goldstein, A. H.: Coupling of organic and inorganic aerosol systems and the effect on gas–particle partitioning in the southeastern US, Atmos. Chem. Phys., 18, 357-370, 10.5194/acp-18-357-2018, 2018.

Rindelaub, J. D., Craig, R. L., Nandy, L., Bondy, A. L., Dutcher, C. S., Shepson, P. B., and Ault, A. P.: Direct Measurement of pH in Individual Particles via Raman Microspectroscopy and Variation in Acidity with Relative Humidity, The Journal of Physical Chemistry A, 120, 911-917, 10.1021/acs.jpca.5b12699, 2016.

Robinson, R. A., and Stokes, R. H.: Electrolyte solutions, Courier Corporation, 2002.

Song, S., Gao, M., Xu, W., Shao, J., Shi, G., Wang, S., Wang, Y., Sun, Y., and McElroy, M. B.: Fine-particle pH for Beijing winter haze as inferred from different thermodynamic equilibrium models, Atmos. Chem. Phys., 18, 7423-7438, 10.5194/acp-18-7423-2018, 2018.

van Boekel, M. A. J. S.: Kinetic Modeling of Reactions In Foods, CRC Press, New York, 2008.

Zünd, A.: Modelling the thermodynamics of mixed organic-inorganic aerosols to predict water activities and phase separations, ETH Zurich, 2007.

---

## Author Response (AR1)

To                                                                          Date:   30th   June 2018
Rob MacKenzie,
Co-editor,
Atmospheric Chemistry and Physics.

**Sub: Submission of revised research article ACP-2018-85 for publication.**

Dear Prof. MacKenzie,

First of all, I would like to thank you on behalf of my co-authors for kindly agreeing to extend the deadline for submission of our revised article. I am hereby submitting the revised version of our manuscript ACP-2018-85. We have revised the manuscript in accordance with the comments and suggestions made by the reviewers.

We have confirmed the mistakes in Eqs. 4 and 9 in the original manuscript as shown below (as we have communicated with Prof. Nenes before). The density of solvent was taken as the density of hygroscopic aerosol by mistake in the original manuscript as details shown in the table below.

| Eq. # | Original (with mistake) | Corrected |
|---|---|---|
| 4 | $y_H = f_H \dfrac{\rho_{\text{sln}}}{\rho_{\text{sln}} + 0.001[M_s\sum c_i - \sum c_i M_i]}$ | $y_H = f_H \dfrac{\rho_{solvent}}{\rho_{\text{sln}} + 0.001[M_s\sum c_i - \sum c_i M_i]}$ |
| 9 | $pH_x - pH_c = -\lg\dfrac{f_H x_H}{c_H y_H} = -\lg\dfrac{0.001Ms}{\rho_{\text{sln}}} = 1.74 + \lg\rho_{\text{sln}}$ | $pH_x - pH_c = -\lg\dfrac{f_H x_H}{c_H y_H} = -\lg\dfrac{0.001Ms}{\rho_{solvent}} = 1.74 + \lg\rho_{solvent}$ |

However, as Dr. Zuend suggested to use more general equations (considering solvents other than water), the corrected equations do not directly appear in the revised manuscript. Instead they have been shown in a more general form as: $f_H = y_H 1000\frac{dm^3}{m^3}\frac{M_w}{\rho_0}\frac{c_H}{x_H}$     (6)     and     $pH_x - pH_c = \log10\frac{1000dm^3/m^3 M_s c^0}{\rho_0}$ (12) in the revised manuscript.

In the original manuscript, we had discussed the effect of the density of aerosol solution on the conversion between molality-based pH and molarity-based pH, which formed the major part of the *Results and Discussion* section. However, after correction, the conversion between molarity and molality-based pH actually does not depend on the density of the solution, but does so on the density of the pure solvent. Therefore the orignal discussion is only partially valid when the acitvity coefficent of H+ is simplified as unity (e.g. by ISORROPIA-II).  Consequently, we have revised the mansucript significantly. The major revisons include: (1) removing the original discussion regarding comparision between pHm and pHc; (2) incorporating the calculation of pH on differnet scales using multiple thermodynamic models;  3) using a gas+aerosol system instead of an aerosol only system to estimate pH; and (4) inclusion of a sub-section to discuss the general issues regarding aerosl acidity comparision across studies. We believe that the importance of our study and the major conclusion remain unaffected by the mistake although a significant revision has been done.

I am also submitting herewith itemized responses to each of the reviewers' comments. All changes made in the manuscript have been highlighted in red fonts.

I sincerely hope that the manuscript in its revised form will satisfy all queries of the reviewers.

With regards,

Wang Xuemei

**Response to Dr. Zuend**

*Comments are in black, responses in blue and the revised text in red.*

**1 General comments**

Acidity is an important characteristic of liquid aerosol particle phases, which often tend to be highly acidic, as expressed by a low pH value. This technical note addresses several issues arising from the existence of distinct ways to define "pH". The authors discuss the differences between pH defined on molality, molarity and mole fractions scales and the importance of using thermodynamically correct conversions among these scales when field data is compared. This scale-dependence of pH is an important point indeed. While this reviewer has been aware of the pH scale dependence and its potential pitfalls for a while, it is an issue appropriate for a "technical note" outlining the proper thermodynamic scale conversion (theory) and providing discussion of related practical issues in aerosol acidity evaluations from field data.

However, the present manuscript contains a number of flaws, several of which are further discussed under specific comments below.

The interactive comment by Simon Clegg (https://www.atmos-chem-phys-discuss.net/acp-2018-85/acp-2018-85-SC1.pdf) summarizes the main concerns shared by this reviewer. Major revisions are necessary to transform this manuscript into a paper that contains (i) a thorough discussion of the thermodynamic scale conversions as well as (ii) a discussion of the general issues with pH estimation of field aerosol samples.

While the latter point is not the focus of this technical note, mostly ignoring the issues of that point is not appropriate either. Errors from incorrect $H^+$ concentration estimations, e.g., by use of an ion charge balance, as done in this study (and others), may frequently be more substantial than the errors from pH value comparisons without proper pH scale conversion. Therefore, a discussion of issues with aerosol acidity determination and pH scale intercomparison must include both.

**Response:**

We thank Dr. Zuend for the constructive comments on our manuscript. We also thank him for agreeing with the importance of pH interconversion between different scales, which is the main focus of the current study.

First of all, we would like to apologize for the mistakes made in Eqs. 4 and 9 in the original manuscript that stem from errors originally made in the equation by van Boekel (2008) used in our reference. The density of solvent was taken as the density of hygroscopic aerosol by mistake in the original manuscript. Since the mistakes are not included in either of the reviewer's comments or in the short comments by Prof. Clegg, we clarify this issue first before responding to the reviewers' comments in detail. The details of the mistakes are shown in Table 1 below.

Table 1. Correction of Eqs. 4 and 9 in the original manuscript

| Eq. # | Original (with mistake) | Corrected [a] |
|---|---|---|
| 4 | $y_H = f_H \dfrac{\rho_{\text{sln}}}{\rho_{\text{sln}} + 0.001[M_s\sum c_i - \sum c_i M_i]}$ | $y_H = f_H \dfrac{\rho_{solvent}}{\rho_{\text{sln}} + 0.001[M_s\sum c_i - \sum c_i M_i]}$ |
| 9 | $pH_x - pH_c = -\lg\dfrac{f_H x_H}{c_H y_H} = -\lg\dfrac{0.001 Ms}{\rho_{\text{sln}}}$ $= 1.74 + \lg\rho_{\text{sln}}$ | $pH_x - pH_c = -\lg\dfrac{f_H x_H}{c_H y_H} = -\lg\dfrac{0.001 Ms}{\rho_{solvent}}$ $= 1.74 + \lg\rho_{solvent}$ |

[a] Since we have taken Dr. Zuend's suggestion to use more genearl equations (considering solvents other than water), the corrected equations do not direcely appear in the revised mansucrit. Instead they have been shown in a more general form as: $f_H = y_H 1000 \dfrac{dm^3}{m^3} \dfrac{M_w}{\rho_0} \dfrac{c_H}{x_H}$ (6) and $pH_x - pH_c = \log 10 \dfrac{1000 dm^3/m^3 M_s c^0}{\rho_0}$ (12) in the revised manuscript.

In the original manuscript, we discussed the effect of the density of aerosol solution on the conversion between molality pH and molarity-based pH, which formed the major part of the *Results and Discussion* section. However, after correction, the conversion between molality and molarity-based pH actually does not depend on the density of the solution but does so on the density of the pure solvent. Therefore the orignal discussion is only partially valid when the acitvity coefficent of $H^+$ is simplified as unity (e.g. by ISORROPIA-II). Consequently, we have revised the mansucript significantly. The major revisons include (1) removing the original discussion regarding comparision between $pH_x$ and $pH_c$; (2) incorporating the calculation of pH on differnet scales using multiple thermodynamci models; 3) using a gas+aerosol system instead of an aerosol only system to estimate pH; 4) discussing the differnce between different scales for a generalized solvent system instead of only water and (5) inclusion of a sub-section to discuss the general issues regarding aerosl acidity comparision across studies. We believe that the importance of our study and the major conclusion remain unaffected by the mistake although a significant revision has been done. Below are our detailed responses to each of the comments.

**Response to general comments:**
We have addressed Prof. Simon Clegg's comments point by point. A major revision of the manuscript has been conducted according to all comments. Especially, we have now revised the method to estimate aerosol acidity using a gas + aerosol system instead of the aerosol only system utilized in original version. We have also added an entire section (Section 3.2) in the revised manuscript to summarize the general issues for pH comparison across studies. A detailed response to each of the Reviewer's comments is listed below:

**2 Specific comments**
1. Abstract, line 16: Stating that this issue is addressed "for the first time" is rather bold given that the theoretical framework for activity coefficient and pH scale con- versions has been known for decades (even though it may be true that it is a frequently ignored issue, therefore it is worthy of attention by the atmospheric chemistry community).

**Response:**
We agree that the description here is inappropriate. We have deleted the concerned phrase in the description (Page 1, Lines 15-18).

**Revised text:**
Such comparisons are however routinely performed in the atmospheric science community. This study attempts to address this issue by comparing $PM_{2.5}$ aerosol pH based on different scales (molarity, molality and mole fraction) on the basis of theoretical considerations followed with a set of field data from Guangzhou, China as an example.

2.  Abstract, line 18: "Using hourly ionic species measurements in Guangzhou, China, it is observed that $pH_x$ (mole fraction based) is always 1.74 pH unit higher than $pH_m$ (molality based)". This is clearly misleading, since the pH unit difference is not truly based on observation. It is a circular argument also made at other places in the manuscript. The 1.74 pH unit difference is in fact coming from the application of such a theory-based difference and using the same thermodynamic model to compute the pH values in different scales based on $H^+$ activity coefficient prediction from the same samples.

    Response:
    We agree with the reviewer's point that the relationship between $pH_x$ and $pH_m$ is from theory. The field data shown in this study is only an example of the application of the theory. The text has been revised as below (Page 1, Lines 16-21).

    **Revised text:**
    This study attempts to address this issue by comparing $PM_{2.5}$ aerosol pH based on different scales (molarity, molality and mole fraction) on the basis of theoretical considerations followed with a set of field data from Guangzhou, China as an example. The three most widely used thermodynamic models (E-AIM-IV, ISORROPIA-II, and AIOMFAC) are employed for the comparison. It has been shown theoretically that the difference between $pH_x$ (mole fraction based) and $pH_m$ (molality based) is always a constant (1.74, when the solvent is water) within a thermodynamic model regardless of aerosol property.

3.  Page 2, line 9: The authors state; "The acidity of aerosols can be quantified by parameters such as strong acidity, free acidity, cation-to-anion ratio and ammonium- to-sulfate ratio. However, these parameters neglect the effect of liquid water content or the dissociation of ions and acids (Pathak et al., 2004; Hennigan et al.,2015)." First, a definition of what strong acidity and free acidity represents is not given. Second, it is incorrect that free acidity neglects water content, as the partial dissociation of species like bisulfate is very much dependent on water content and therefore affects free acidity. Pathak et al. and Hennigan et al. do not seem to make such a point.

    Response:
    We agree that our original description was vague and inaccurate. In the revised manuscript, we have (1) added the definition of both strong and free acidity; and (2) specified the drawbacks of these parameters with proper citation (Page 2, Lines 17-26).

    **Revised text:**

The most accurate parameter to characterize aerosol acidity is considered to be pH. The other parameters often used as proxies of aerosol acidity do not offer information on how acidic the particles are when they are present as aqueous droplets (Pathak et al., 2004). For example, strong acidity (defined as nmol of total $H^+$ per $m^3$ of air measured in water extracts of particles using the USEPA Reference Method (USEPA, 1992)) and ion charge balance are unable to distinguish between free and undissociated $H^+$ (e.g., protons associated with bisulfate) (Pathak et al., 2004;Hennigan et al., 2015). Ammonium-to-sulfate ratio and cation-to-anion ratio are unable to provide any measure of the degree of aerosol acidity even qualitatively (Hennigan et al., 2015). And lastly, free acidity (defined as the actual concentration of free $H^+$ per $m^3$ of air, not including the $H^+$ released from bisulfate ions in aqueous extracts) represents the quantity of $H^+$ in a specific volume of air while neglecting the concentration of $H^+$ in liquid water (Pathak et al., 2004).

4. Equation (1): Define the meaning of "lg".

Response:
We have revised lg to log10 to make it clearer throughout the manuscript.

5. Equation (2): This equation and its description is flawed. (1) What are the terms of "1000" in the numerator and denominator? The authors likely use these for conversion from units of kg to g. If so, the mathematically correct way of writing this would be to write $1000 \frac{g}{kg}$ in the equation and it would be necessary to state that the molar masses are supposed to be used in units of $\frac{g}{mol}$ rather than the standard SI unit of $\frac{kg}{mol}$. Otherwise it is simply incorrect and a potential source of confusion. However, since the ACP recommendation is to use SI units whenever possible, there is not need for these unit conversion terms at all (they would be 1). (2) This expression is only correct for the special case where the only solvent for the ions is water. However, in the more general case, there may be other solvents, such as organic compounds mixed with water and the ions in a liquid phase. In that case, the distinct molar masses of the organics must be accounted for in the activity coefficient conversion expression. Hence, since this is a key part of the discussion about different thermodynamic composition scales and reference and standard states, it should be shown correctly for the general case. A rigorous derivation of such scale conversions is, e.g., shown in the PhD thesis by Zuend (2007) (page 45 – 47 there), which shows different versions of the scale conversion formula. One of which (useful here) is $\ln[\gamma_i] = \ln[f_i^*] + \ln[\frac{x_i}{m_i M_w}]$ and f* the mole fraction scale activity coefficient, both with reference state of infinite dilution in pure water; $x_i$ the ion mole fraction with respect to dissociated ions, $m_i$ the ion molality and $M_w$ the molar mass of water. Since solvents other than water are also included in both the mole fraction and molality expressions for ion "i", this is a general expression. From this scale conversion of ion activity coefficients, it is readily shown that the difference in pH values is generally given by $pH_x - pH_m = -\log10[m^0 M_w]$, where m° denotes unit molality (= 1 mol/kg) (similar to Eq. (8) in the manuscript, but note the difference in units, the given Eq. (8) is not dimensionless in the log).

Response:
We thank the Reviewer for pointing out this issue.
   1) The factor of 1000 in the original equation was for the conversion of kg to g. The unit of molecular mass was g/mol in the original manuscript. We have now revised the unit of

molecular mass to the SI unit of kg mol$^{-1}$ as suggested. All other units in the manuscript are now SI units except molarity (mol dm$^{-3}$). So, $1000\frac{dm^3}{m^3}$ as a conversation factor is still shown when molarity of ion is involved.

2) We agree that the original Eq.2 is only valid when water is the only solvent. We have now cited Eq. 2.59 in the PhD thesis by Zünd (2007) and Robinson and Stokes (2002) to show the conversion (1) between $\gamma_H$ and $f_H$ (Eq. 4); (2) between $\gamma_H$ and $y_H$ (Eq. 5); and (3) between $f_H$ and $y_H$ (Eq. 6) in a more general form as below (Table 2).

**Revised equations:**

$$\gamma_H = f_H \frac{x_H}{m_H M_s} \tag{4}$$

$$\gamma_H = 1000 \frac{dm^3}{m^3} \frac{c_H y_H}{m_H \rho_0} \tag{5}$$

$$f_H = y_H 1000 \frac{dm^3}{m^3} \frac{M_w}{\rho_0} \frac{c_H}{x_H} \tag{6}$$

3) For the interconversion of concentrations in different scales, we have used the parameter $M_s$ to represent the molar mass of the solvent (for single solvent) or the averaged molar mass for the scenario of multiple solvents in the system (Table 2).

**Revised equations:**

$$x_H = \frac{m_H}{\sum m_i + \frac{1}{M_S}} \tag{7}$$

$$c_H = \frac{m_H}{\frac{\sum m_i M_i + 1}{\rho_{sln}}} \tag{8}$$

$$x_H = \frac{M_S c_H}{M_S \sum c_i + 0.001 \frac{m^3}{dm^3} \rho_{sln} - \sum c_i M_i} \tag{9}$$

6. As pointed out in the comment by Simon Clegg, thermodynamic models differ in the way single-ion activity coefficients are calculated (since only mean cation/anion pair activity coefficients are measureable) and of course they also differ in the expressions, such that even with the correct conversion of activity coefficients or pH values between scales, different models may predict different $pH_m$ (or $pH_x$) values for the same input composition. Furthermore, only some models account for the influence of organic species in the mixture (e.g. AIOMFAC can be used for that, while ISORROPIA is only for inorganic aqueous mixtures) and differences in predicted pH may partially stem from organic interactions with ions and treatment of phase separation, see Pye et al (2018).

**Response:**
We agree with the Reviewer's point. This comment has been handled in combination with #10. We have now added Section 3.2 to discuss the general issues when comparing aerosol acidity across studies (including the difference in models, forward vs. reverse mode, stable vs. metastable mode, effect of non-volatile cations and effect of organic compounds) (Page 8, Line 12 to Page 9, Line 14).

**Revised text:**

3.2 General issues with pH comparison

It has been shown above that proper scale conversion has to be conducted when aerosol pH is compared. However, one should bear in mind that even with the same measured data and scale, pH calculated with different thermodynamic models or with different parameters may still not be comparable. Below, we briefly describe some of the general issues that need to be considered when aerosol acidity is compared across studies along with a summary of parameters used in the published studies in Table S1.

(1) Thermodynamic models like ISORROPIA-II and E-AIM can run in forward mode and reverse mode which result in significant difference (Song et al., 2018;Hennigan et al., 2015). It is recommended to use thermodynamic models in forward mode (gas plus aerosol as input) instead of reverse mode (aerosol only as input) which is highly sensitive to measurement uncertainties (Hennigan et al., 2015).

(2) Thermodynamic model can also be run in stable (liquid only) or metastable modes (both solid and liquid) which has not been specified in many studies (Table S1). Song et al. (2018) have shown that model calculations coupled with stable or metastable state assumptions can provide reasonable estimates of aerosol water and pH. However, as pointed by Song et al. (2018), the studies using standard ISORROPIA-II (without code correction) running in stable mode have predicted unrealistic pH values of around 7 and should be reevaluated.

(3) The effect of non-volatile cations such as $Na^+$, $Ca^{2+}$, $Mg^{2+}$ and $K^+$ on aerosol pH may also not be ignored. Although the effect of non-volatile cations on pH may be only 0.2-0.5 pH units, their impact on predicted partitioning of a semi-volatile species can be significant due to the highly non-linear response of $NH_3$-$NH_4^+$ or $HNO_3$-$NO_3^-$ partitioning to pH (Guo et al., 2017). Since E-AIM cannot explicitly treat $Ca^{2+}$, $Mg^{2+}$ and $K^+$ (unlike ISORROPIA-II and AIOMFAC), pH estimated using E-AIM may ignore $Ca^{2+}$, $Mg^{2+}$ and $K^+$ (as shown in Table S1) or treat them as equivalent sodium (e.g. (Hennigan et al., 2015)). Even if all non-volatile cations are treated as $Na^+$, the predicted thermodynamic states can be biased due to the strong non-ideality of divalent ions as well as variations in water uptake characteristics between $Na^+$ salts and its counterparts (e.g., Fountoukis et al., 2009).

(4) Most studies so far have estimated pH of aerosols with only inorganic compounds. However, it has been reported that pH can be affected by organic compounds in several ways. Guo et al., (2015) have shown that the pH can be increased by 0.15 to 0.23 units when aerosol water associated with organic compounds is considered. Omission of the contribution of organic acids to $H^+$ has been estimated to increase the pH by 0.07±0.03 by Song et al. (2018) using E-AIM-IV. It has been shown recently that accounting for non-ideal mixing can modify the pH such that a fully interactive inorganic–organic system showed a pH roughly 0.7 units higher than that predicted using an inorganic only system by AIOMFAC (Pye et al., 2018).

7. Equation (4): Similar to above comment. A general expression should be shown, with proper use of units.

**Response:**

Thanks. Equation 4 has been revised accordingly (Eq. 6 in the revised manuscript).

**Revised equations:**

$$f_H = y_H 1000 \frac{dm^3}{m^3} \frac{M_w}{\rho_0} \frac{c_H}{x_H} \qquad (6)$$

8. Page 4, line 3: the last sentence there makes little sense. The pH values can be compared when the scale effect is accounted for; the point is that one should not expect the values to be equal.

   **Response:**
   We assume that the Reviewer actually meant page 5 in the original manuscript. This sentence has been deleted in the revised manuscript.

9. Page 4, line 13: "This is supported by our field data". Again, this is a circular argument. The $pH_x - pH_m$ scale difference is used in the evaluation of the pH values, so of course it will show as consistent, but the measurements are no proof for that. Also, the fact that the difference should be a constant in pH units is clear from the theory, as long as the same thermodynamic model is used to compute the activity coefficients (which may not be the case when different studies are compared).

   **Response:**
   We assume the Reviewer actually meant page 5 in the original manuscript. We agree with the reviewer's point and have revised the text accordingly. As we clarified in the beginning of our response, other revisions have also been made and the final revised text is as below (Page 6, Lines 24-27).

   **Revised text:**
   The difference of $pH_x$ and $pH_m$ is $\log 10 M_s m^0$ (according to Eq. (10)) which is only determined by the molecular weight of the solvent. When water is the only solvent in the system (molecular weight of 0.018 kg mol$^{-1}$), $pH_x - pH_m$ is fixed at 1.74 within the model regardless of aerosol property or the model (as in this study).

10. Page 7, line 6: The discussion in this paragraph is not sufficient to address the other very important issues when aerosol acidity is attempted to be assessed from field measurements. It is also clear from theory and comparisons that organic compounds will affect aerosol acidity, maybe not dramatically but noticeably, since their interactions with water and $H^+$ ions are affecting the activity of $H^+$. Last sentence in paragraph: "The relationship between pHx, pHc and pHm established in this study is valid regardless of the method selected to estimate aerosol acidity." This is true only for the scale conversion, since it depends on proper application of thermodynamic theory only (which has been known for decades and is not a novelty of this study). However, when aerosol sample acidity is estimated in practice, different models are used for activity coefficients (e.g. ISORROPIA, E-AIM, AIOMFAC) or unit activity coefficients are assumed (not recommended). Moreover, different methods are applied to determine the approximate H+ amount, which is a critical problem in acidity evaluation, see Hennigan et al (2015). Therefore, the difference in reported pH values is not just due to the offsets between these different pH scales. The authors have failed to make this important point very clear.

**Response:**

We agree with the reviewer's point. This comment has been handled in combination with #6. We have now added Section 3.2 to discuss the general issues when comparing aerosol acidity across studies (including difference in models, forward vs. reverse mode, stable vs. metastable mode, effect of non-volatile cations and effect of organic compounds). The revised text is shown in the response to comment #6.

11. Table 1: The definitions include many mistakes and typos; e.g., $pH_m$ is not molarity based, the last two entries are confusing and not correctly described and reference states of activity coefficients are missing.

**Response:**

Table 1 has been revised thoroughly. The reference state of activity and activity coefficient have been added in the definition of activity. The revised Table 1 is shown below.

**Revised table:**

**Table 1. List of abbreviations.**

[revised manuscript text omitted]

**Response to Anonymous Referee #2**

*Comments are in black, responses in blue and the revised text in red.*

Particle acidity is an important property in terms of aerosol chemistry and its impact on health and climate. Due to the importance, there has been an increasing amount of publication investigating particle pH through thermodynamic models in recent years, which is a step forward compared to the problematic ion balance or molar ratio because of particle liquid water and non-ideality effect (activity coefficient). Without a consensus on the pH definition, it would be difficult to compare various studies, and this may lead to misunderstanding or misinterpretation of particle pH. Although some particle acidity studies state clearly the pH definition used in the analyses, there are some papers not talking about the definition at all. In the latter case, some studies probably take the approach of molality-based or molarity-based because they take a pH of 7 as neutral.

This paper shows the two pHs are very close and the minor difference is caused by particle density (for dilute water solution, no difference is expected from the two pHs). In general, this paper hits an important point (i.e., pH definition) that has not been paid enough attention to by the community and fits the scope of a technical note on ACP. However, some revisions are needed before considering a publication.

**Response:**
We thank the Reviewer for the helpful comments and suggestions, and also for agreeing with the importance of pH interconversion between different scales, which is the main focus of current study.

First of all, we would like to apologize for the mistakes made in Eqs. 4 and 9 in the original manuscript that stem from errors originally made in the equation by van Boekel (2008) used in our reference. The density of solvent was taken as the density of hygroscopic aerosol by mistake in the original manuscript. Since the mistakes are not included in either of the reviewer's comments or in the short comments by Prof. Clegg, we clarify this issue first before responding to the reviewers' comments in detail. The details of the mistakes are shown in Table 1 below.

Table 1. Correction of Eqs. 4 and 9 in the original manuscript

| Eq. # | Original (with mistake) | Corrected [a] |
|---|---|---|
| 4 | $y_H = f_H \dfrac{\rho_{\text{sln}}}{\rho_{\text{sln}} + 0.001[M_s\sum c_i - \sum c_i M_i]}$ | $y_H = f_H \dfrac{\rho_{solvent}}{\rho_{\text{sln}} + 0.001[M_s\sum c_i - \sum c_i M_i]}$ |
| 9 | $pH_x - pH_c = -\lg \dfrac{f_H x_H}{c_H y_H} = -\lg \dfrac{0.001 Ms}{\rho_{\text{sln}}}$ $= 1.74 + \lg\rho_{\text{sln}}$ | $pH_x - pH_c = -\lg \dfrac{f_H x_H}{c_H y_H} = -\lg \dfrac{0.001 Ms}{\rho_{solvent}}$ $= 1.74 + \lg\rho_{solvent}$ |

[a] Since Dr. Zuend suggested to use more genearl equations (considering solvents other than water), the corrected equations do not direcely appear in the revised mansucript. Instead they have been shown in a more general form as: $f_H = y_H 1000 \dfrac{dm^3}{m^3} \dfrac{M_w}{\rho_0} \dfrac{c_H}{x_H}$ (6) and $pH_x - pH_c = \log 10 \dfrac{1000 dm^3/m^3 M_s c^0}{\rho_0}$ (12) in the revised manuscript.

In the original manuscript, we discussed the effect of the density of aerosol solution on the conversion between molality-based pH and molarity-based pH, which formed the major part of the *Results and Discussion* section. However, after correction, the conversion between molarity and molality-based pH actually does not depend on the density of the solution but does so on the density of the pure solvent. Therefore the orignal discussion is only partially valid when the acitvity coefficent of H⁺ is simplified as unity (e.g. by ISORROPIA-II). Consequently, we have revised the mansucript significantly. The major revisons include (1) removing the original discussion regarding comparision between $pH_m$ and $pH_c$; (2) incorporating the calculation of pH on differnet scales using multiple thermodynamci models;  3) using a gas+aerosol system instead of an aerosol only system to estimate pH; 4) discussing the differnce between different scales for a generalized solvent system instead of only water and (5) inclusion of a sub-section to discuss the general issues regarding aerosl acidity comparision across studies. We believe that the importance of our study and the major conclusion remain unaffected by the mistake although a significant revision has been done. Below are our detailed responses to each of the comments.

1. There are several "circular" statements as pointed out by the other reviewer Andreas Zuend. The differences between pHx, pHc, pHm are all expected from their definitions. This is the key point of this paper:  the difference between pHc and pHm is small (within 0.2 pH units, caused by particle density), but pHx is significantly higher than pHc and pHm (pHx – pHm = 1.74).  The E-AIM (or any other model) predicted pHs are supposed to be consistent with the rule, if one model is used consistently in this analysis.  Therefore, it is not a real support by field data (used as E-AIM input) as claimed on line 13 Page 5.  Relating to this, the author should consider changing the saying of "observed" (Line 18 Page 1) in the abstract. The presented result is all based on a thermodynamic model prediction of pH but not measured pH.

   **Response:**
   We agree with the Reviewer's point. The differences between *pHx*, *pHc* and *pHm* are expected from their definitions. The field data shown in this study is actually an example to show the conversion between different scales. The following revision has been done in the revised manuscript (Page 1, Lines 16-18).

   **Revised text**

   This study attempts to address this issue by comparing $PM_{2.5}$ aerosol pH based on different scales (molarity, molality and mole fraction) on the basis of theoretical considerations followed with a set of field data from Guangzhou, China as an example.

2. From a boarder view of an application, this paper could be more beneficial by showing the default pH scales given by widely used thermodynamic models. The E-AIM model has been discussed in detail, however, ISORROPIA or AIOMFAC or any other model is not mentioned. For example, ISORROPIA gives pH in the model output and the pH scale is molality-based (Fountoukis et al., 2009; Guo et al., 2015). If a literature doesn't specify the pH scale, this piece of info could be very helpful to readers.

**Response:**
We agree with the Reviewer's point.
(1) We have addressed this issue in the *Introduction* section (Page 3, Lines 17-21).

**Revised text:**
It appears that the selection of the standard state of activity is arbitrary for aerosol acidity studies, and is not always defined in published articles when pH is used to characterize the acidity of aerosol (8 out of 32 studies as shown in Table S1). This may not be problematic in the case of ISORROPIA-II where the default output pH is always molality-based; however, confusion is possible when E-AIM or AIOMFAC are used since these models provide both molality- and mole fraction-based concentrations as output.

(2) Following the reviewer's suggestion, we have now shown the calculations with all the 3 models. We have also described the parameters that can be obtained from the model outputs as well as how other parameters are estimated. The revised text is shown below (Table S2).

**Revised text:**

Table S2. A summary of estimation methods of parameters for pH calculation based on different standard states.

| Parameters | E-AIM-IV | ISORROPIA-II | AIOMFAC |
|---|---|---|---|
| **Mole fraction** | | | |
| $x_H$ | output | Eq. (7) | output |
| $f_H$ | output | 1[a] | Eq. (4) |
| **Molality** | | | |
| $m_H$ | output | output | output |
| $\gamma_H$ | Eq. (4) | 1[a] | output |
| **Molarity** | | | |
| $c_H$ | Eq. (8)[b] | Eq. (8)[b] | Eq. (8)[b] |
| $y_H$ | Eq. (6) | 1[a] | Eq. (6) |

Note: [a] activity coefficient is assumed to be 1; [b] the density of aerosol solution is based on the result from E-AIM-IV.

3. The section 3.3 is problematic without discussion of uncertainty, especially considering that the presented pH in this study is solely based on particle data (no gas data used to constrain pH). The predicted pH uncertainty is propagated from the particle ionic composition data (6% reported by Chen et al. (2016)) and RH, T. It is not easy to estimate particle pH uncertainty. Guo et al. (2015) estimated a pH uncertainty of 13% using another model, ISORROPIA, and in forward model for their dataset. Even though the two data points (S-I and S-II) are selected for the largest deviation from the 1:1 line, small differences (7% or 8%) in hydrogen ion activity are found, which seem to be within uncertainty range. Hennigan et al. (2015) and other papers have pointed out that forward mode is superior over reverse mode in terms of particle pH prediction accuracy. For one reason, reverse mode is more sensitive to particle measurement uncertainty (likely the cause for occasionally very high pH seen in Figure 1). For the other reason, particle pH is sensitive to gas-particle partitioning of semivolatile species (e.g., NH4+, NO3-, and Cl-), as long as the species is not totally in gas phase or particle

phase. The forward mode predicted pH can then be validated if predicted and observed gas-particle partitioning agree. In contrast, the reverse mode predicted pH could not be verified in the same way. The Line 7 on Page 7 seems to suggest there is no advantage of using forward mode calculation and this is misleading to potential users.

**Response:**
We thank the reviewer for pointing out this important issue.
(1) We have revised the method to calculate aerosol pH from the forward mode, which is of lower uncertainty, instead of reverse mode.
(2) We acknowledge that the difference between $pH_m$ and $pH_c$ may not be large compared with uncertainties from measurement errors. Therefore, to address the limitation of our finding, we have added the following text in the discussion (Page 7, Line 29 to Page 8, Line 5).

**Revised text:**
Given that the uncertainty of pH calculation due to measurement errors can be as high as 14% (Guo et al., 2015), the difference of $pH_c$ and $pH_m$ can simply fall within the range of measurement errors. However, the bias between $pH_c$ and $pH_m$ can be considered to be a systematic one, which needs to be addressed for the sake of comprehensiveness in theoretical analysis. Moreover, even small biases in pH may imply substantial partitioning errors for semivolatile species like ammonium, nitrate, chloride, and even organic acids (Guo et al., 2017).

4. As Andreas Zuend points out, it would be nice to mention the possible differences be- tween predicted pH via different models at the end of section 3.4. Even if the same inputs are given to models, 100% agreement in pH is not expected due to differences in assumptions and approaches. There are a number of studies comparing thermodynamic models and exploring the differences (Hennigan et al., 2015; Liu et al., 2017; Pye et al., 2018; Song et al., 2018)

**Response:**
We agree with the Reviewer's point. We have addressed this issue in the revised manuscript.

(1) Following the reviewer's suggestion, we have shown the comparison of pH calculated using different thermodynamic models in the revised manuscript as below followed by a comparison of the three models (Table 3 and Page 6, Lines 2 to 18).

[revised manuscript text omitted]

(3) The effect of non-volatile cations such as $Na^+$, $Ca^{2+}$, $Mg^{2+}$ and $K^+$ on aerosol pH may also not be ignored. Although the effect of non-volatile cations on pH may be only 0.2-0.5 pH units, their impact on predicted partitioning of a semi-volatile species can be significant due to the highly non-linear response of $NH_3$-$NH_4^+$ or $HNO_3$-$NO_3^-$ partitioning to pH (Guo et al., 2017). Since E-AIM cannot explicitly treat $Ca^{2+}$, $Mg^{2+}$ and $K^+$ (unlike ISORROPIA-II and AIOMFAC), pH estimated using E-AIM may ignore $Ca^{2+}$, $Mg^{2+}$ and $K^+$ (as shown in Table S1) or treat them as equivalent sodium (e.g. (Hennigan et al., 2015)). Even if all non-volatile cations are treated as $Na^+$, the predicted thermodynamic states can be biased due to the strong non-ideality of divalent ions as well as variations in water uptake characteristics between $Na^+$ salts and its counterparts (e.g., Fountoukis et al., 2009).

(4) Most studies so far have estimated pH of aerosols with only inorganic compounds. However, it has been reported that pH can be affected by organic compounds in several ways. Guo et al., (2015) have shown that the pH can be increased by 0.15 to 0.23 units when aerosol water associated with organic compounds is considered. Omission of the contribution of organic acids to $H^+$ has been estimated to increase the pH by 0.07±0.03 by Song et al. (2018) using E-AIM-IV. It has been shown recently that accounting for non-ideal mixing can modify the pH such that a fully interactive inorganic–organic system showed a pH roughly 0.7 units higher than that predicted using an inorganic only system

by AIOMFAC (Pye et al., 2018).

**Minor comments:**

1. Toning down the statement of "for the first time" is suggested. Thermodynamic specialists must have known the difference between pH scales. The elucidation in the paper is based on established equations on textbooks and doesn't sound to me like a groundbreaking finding. The paper is of value to minimize the gaps in the current understanding of pH definitions.

**Response:**
Thanks for pointing out the issue. We have deleted the concerned phrase in the description and elsewhere in the revised manuscript (Page 1, Lines 16-18).

**Revised text:**
This study attempts to address this issue by comparing $PM_{2.5}$ aerosol pH based on different scales (molarity, molality and mole fraction) on the basis of theoretical considerations followed with a set of field data from Guangzhou, China as an example.

2. The units in Equation (3) (mol/kg water) and (6) (mol/dm3) look redundant. However, more explanations in the text are needed. Equation (3) is defined based on the mass of water, while Equation (6) is defined based on the volume of particle, which includes the volumes of water (solvent) and other solutes. If Equation (6) is defined solely on the volume of particle water, there would not be any difference with Equation (3). This is not clear in the text.

**Response:**
We thank the reviewer pointing out the issue. The two equations have been changed accordingly (Page 5, Lines 16-17). We have also explained that molarity means mol $dm^{-3}$ solution while molality means mol $kg^{-1}$ solvent in Table 1.

**Revise equations:**

$$pH_c = -\log10(a_{c_H}) = -\log10(\frac{y_H c_H}{c^o}) \tag{2}$$

$$pH_m = -\log10(a_{m_H}) = -\log10(\frac{\gamma_H m_H}{m^o}) \tag{3}$$

**Explanation of parameters (in Table 1):**
$m_H$        molality of hydrogen ions (mol $kg^{-1}$ solvent)
$c_H$        molarity of hydrogen ion (mol $dm^{-3}$ solution)

3. Proper reference as suggested by Simon Clegg.

**Response:**
We have now added the sources of all equations as suggested by Prof. Simon Clegg in Table 2.

**Revised text:**

[revised manuscript text omitted]

---

## Author Response (AR2)

To                                                                           Date: 8$^{th}$ July 2018
Rob MacKenzie,
Co-editor,
Atmospheric Chemistry and Physics.

**Sub: Submission of revised research article ACP-2018-85 for publication.**

Dear Prof. MacKenzie,

I would like to thank you on behalf of my co-authors for your comments on our manuscript. I am hereby submitting the revised version of our manuscript ACP-2018-85. We have revised it in accordance with your comments. All changes made have been highlighted in red fonts in the manuscript as details listed below. The original comments by you are given in black text and our responses is given in blue text.

1. Throughout: please replace "log10" with "log_{10}" - i.e., subscript the base of the logarithm function.
   Response:
   All "log10" have been replaced with "log$_{10}$" throughout the manuscript including Page 5 (Line 15-17), Page 6 (Line 25), Page 7 (Lines 1 and 2), and Page 16 (Table 2).

2. Throughout: I think it is easier to read the values with uncertainties in the form $(1.98\pm2.50)\times10^{-2}$ rather than $1.98\times10^{-2}\pm2.50\times10^{-2}$.
   Response:
   We have revised all similar expressions in the manuscript including Page 6 (Line 10), Page 7 (Line 12) and Page 17 (Table 3).

3. p1, line 20: please replace "It has been shown theoretically" with "Established theory dictates"
   Response:
   We have revised accordingly (Page 1, Line 20).

4. p2, line 27. should be "negative log (base 10) of the activity of..."
   Response:
   We have revised accordingly (Page 2, Line 28).

5. p4, line 2. please write " this issue has not been addressed with sufficient care..."
   Response:
   We have revised accordingly (Page 4, Lines 2-3).

6. p6, line 4. Should be "There are slight differences..."
   Response:
   We have changed "difference" to "differences" in Page 6, Line 4.

7. p6, line 24ff. Please recognise that this result is already in the literature by citing eg Robinson and Stokes.
   Response:
   We have added the citation "e.g. Robinson and Stokes, 2002" in Page 6, Line 24.

I sincerely hope that the manuscript in its revised form will now be accepted for publication.

With regards,

Wang Xuemei

[revised manuscript text omitted]